# The extraordinary boundary transition in the 3d O(N) model via conformal bootstrap

**Jaychandran Padayasi,**[1] **Abijith Krishnan,**[2] **Max A. Metlitski,**[2] **Ilya A. Gruzberg,**[1] **Marco Meineri**[3]

[1] *Department of Physics, The Ohio State University, Columbus, OH 43210, USA*

[2] *Department of Physics, Massachusetts Institute of Technology, Cambridge, MA 02139, USA*

[3] *Département de Physique Théorique, Université de Genève, 24 quai Ernest-Ansermet, 1211 Genève 4, Suisse*

*E-mail:* marco.meineri@gmail.com

ABSTRACT:

This paper studies the critical behavior of the 3d classical $O(N)$ model with a boundary. Recently, one of us established that upon treating $N$ as a continuous variable, there exists a critical value $N_c > 2$ such that for $2 \leq N < N_c$ the model exhibits a new extraordinary-log boundary universality class, if the symmetry preserving interactions on the boundary are enhanced. $N_c$ is determined by a ratio of universal amplitudes in the normal universality class, where instead a symmetry breaking field is applied on the boundary. We study the normal universality class using the numerical conformal bootstrap. We find truncated solutions to the crossing equation that indicate $N_c \approx 5$. Additionally, we use semi-definite programming to place rigorous bounds on the boundary CFT data of interest to conclude that $N_c > 3$, under a certain positivity assumption which we check in various perturbative limits.

# Contents

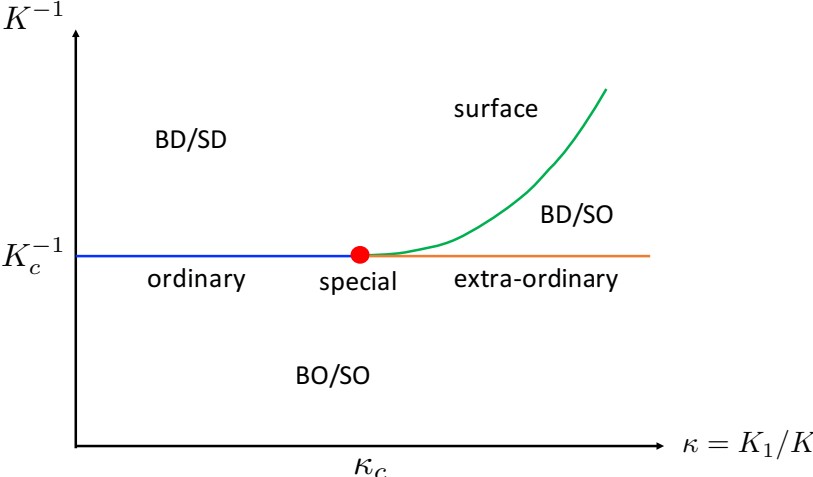

**Figure 1**: Conventionally accepted phase diagram of the classical $O(N)$ model with a boundary in dimension $d > 3$. BO stands for bulk ordered, SO - surface ordered, BD - bulk disordered, SD - surface disordered. For $d = 3$ and $N = 1$ the phase diagram is the same. For $d = 3$ and $N = 2$ the phase diagram has the same topology, but the BD/SO region only has quasi-long-range surface order.

## 1 Introduction

The boundary behavior of systems that are critical in the bulk is a subject with a venerable history [1–3] that has received renewed attention in recent years. The reason for the resurgent interest in boundary criticality is two-fold. First, due to advances in conformal bootstrap, the last decade has seen significant progress in our understanding of conformal field theory (CFT) in dimension $d > 2$. While most of the attention to date has focused on bootstrapping the bulk behavior of CFTs, applications to boundary conformal field theory (BCFT) have also been studied [4–15], the present paper builds on these developments. Second, the discovery of topological insulators and more broadly, advances in understanding topological phases of quantum matter has spurred a wave of interest in boundary behavior in general. Such phases are gapped in the bulk, but may support protected gapless boundary states. While the existence of a bulk gap was originally thought to be crucial for the protection of boundary states, recent work revealed that boundary states may survive in some form even when the bulk gap closes [16–24]. The study of boundary behavior of such gapless bulk systems falls squarely in the domain of boundary criticality. As the boundary behavior of certain quantum spin systems was investigated in this light [25–30], unresolved qualitative questions about boundary criticality in one of the simplest textbook statistical mechanics models—the *classical* $O(N)$ model in $d = 3$—were uncovered. We now discuss what these questions are and how the present paper aims to address them.

Consider the following prototypical lattice realization of the classical $O(N)$ model with a boundary:

$$\frac{H}{k_B T} = -\sum_{\langle ij \rangle} K_{ij} \vec{S}_i \cdot \vec{S}_j. \tag{1.1}$$

Here $\vec{S}_i$ are classical $O(N)$ spins ($N$ component vectors of unit length) at the sites of a semi-infinite $d$-dimensional hypercubic lattice. $K_{ij} > 0$ is a nearest neighbour coupling that is $K_1$ if both $i$ and $j$ belong to the surface layer and $K$ otherwise. Above its lower critical dimension, this model has a

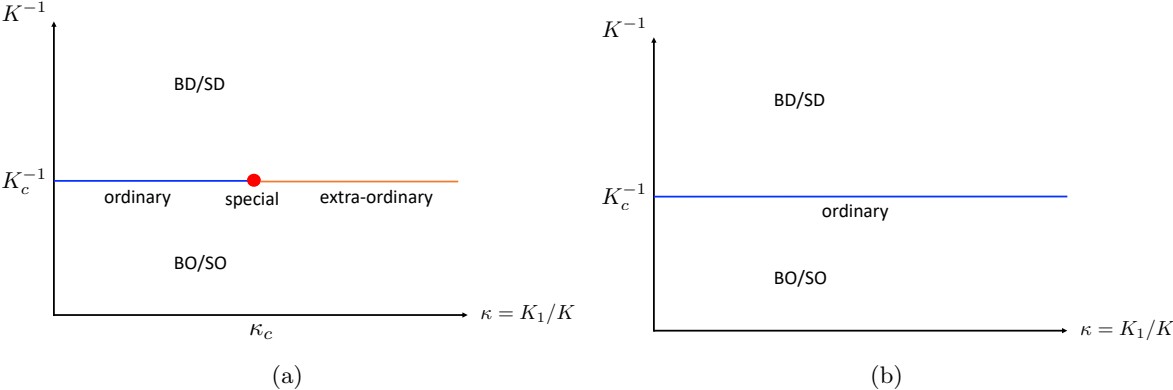

**Figure 2**: Possible boundary phase diagrams of the classical $O(N)$ model for $d = 3$ and $N > 2$. Ref. [31] argued that the phase diagram on the left is realized for $2 < N < N_c$ with the extraordinary phase being of the "extraordinary-log" character, while the phase diagram on the right is realized for large $N$. Here $N_c > 2$ is a yet unknown critical value of $N$. See Ref. [31] for scenarios for the evolution of the phase diagram from one on the left to one on the right with increasing $N$.

bulk phase transition at $K = K_c$. We are specifically interested in the boundary phase diagram of the model (1.1) in bulk dimension $d = 3$ when $N > 2$, which surprisingly is still not settled.

First, we review the situation in dimension $d > 3$ where more clarity exists.[1] For $d > 3$ the conventionally accepted phase diagram has the schematic shape shown in figure 1.[2] Let us define the parameter $\kappa = K_1/K$. For $\kappa$ smaller than a critical value $\kappa_c$, the onset of both the bulk and the boundary order happens at $K = K_c$. This boundary universality class is known as "ordinary". For $\kappa > \kappa_c$ the enhancement of the surface coupling leads to the boundary ordering at a higher temperature than the bulk. Then for $\kappa > \kappa_c$ the onset of bulk order at $K = K_c$ in the presence of established boundary order is known as the "extraordinary" boundary universality class. Finally, the multicritical point at $\kappa = \kappa_c$ and $K = K_c$ is known as the "special" boundary universality class.[3]

Let us now turn our attention to dimension $d = 3$, our main focus in the present paper. For the case of Ising spins the boundary phase diagram remains the same as in figure 1. For $N = 2$, the phase diagram has the same topology as in figure 1, however, now the region labeled as BD/SO has only quasi-long-range boundary order (correlation functions of the boundary order parameter fall off as a power law with a variable exponent) rather than true long range order [32–34]. For $N > 2$, the boundary has a finite correlation length for $K < K_c$ and the only phase transition expected is at $K = K_c$. Thus, the topology of the phase diagram does not mandate the existence of the extraordinary and special boundary universality classes: it is possible that the ordinary universality class is realized for all values of $\kappa$, see figure 2b. However, it is also possible that different boundary universality classes are realized for different values of $\kappa$ even though they would connect the same bulk-disordered/surface-disordered and bulk-ordered/surface-ordered phases, see figure 2a. While such a scenario appears exotic, in Ref. [31] it was argued to be realized for a finite range of $N$: $2 < N < N_c$.

---

[1] Here and below, we often formally treat variables $d$ and $N$ as continuous.

[2] We note, however, that as discussed in Ref. [31] there exists a scenario where the phase diagram in figure 1 needs to be modified in dimensions $3 < d < d_c < 4$.

[3] When we use the term "boundary universality class" here and below we always imply that we are at the bulk critical point $K = K_c$.

Here $N$ is formally treated as a continuous variable and $N_c > 2$ is a yet unknown critical value of $N$.[4] In this range (and also at $N = 2$) the region $\kappa \gg 1$ realizes what was termed the "extraordinary-log" boundary universality class, where the boundary correlation function falls off as

$$\langle \vec{S}_{\vec{x}} \cdot \vec{S}_{\vec{y}} \rangle \sim \frac{1}{(\log |\vec{x} - \vec{y}|)^q}, \tag{1.2}$$

where $q$ is a universal $N$-dependent exponent. Thus, the correlation function of the boundary order parameter falls off extremely slowly: the boundary is almost but not quite ordered. In contrast, Ref. [31] argued that for large $N$ the simple phase diagram in figure 2b is realized. Several scenarios for the evolution of the phase diagram with increasing $N$ from that in figure 2a to that in figure 2b have been discussed in Ref. [31]—we will not attempt to resolve which of these scenarios is realized in this paper.

Recent Monte Carlo simulations of the O($N$) model in $d = 3$ support the above picture. Ref. [35] studied the case $N = 3$ and concluded that the phase diagram in figure 2a is indeed realized. This agrees with the results of an earlier Monte Carlo study, Ref. [36]. Further, the behavior in the extraordinary region found in Ref. [35] appears consistent with the extraordinary-log universality class. In the $N = 2$ model the extraordinary region was recently studied in Ref. [37]; again, results consistent with the extraordinary-log universality class were obtained. Finally, an older study [38] of the $N = 4$ model also obtained the phase diagram in figure 2a. Thus, Monte Carlo results to date indicate that the critical value $N_c$ is very likely greater than three and potentially greater than four.

The central goal of this paper will be to determine $N_c$ using numerical conformal bootstrap. Under certain assumptions, we will be able to place a rigorous bound,

$$N_c > 3. \tag{1.3}$$

In fact, our findings suggest $N_c > 4$, although we cannot make this claim with the same degree of rigour as $N_c > 3$. We now summarize how these results are obtained.

As was shown in Ref. [31], much of the physics of the extraordinary-log universality class including the exponent $q$ in eq. (1.2) and the value of $N_c$ is determined by yet another boundary universality class: the normal universality class. The latter is obtained by applying an explicit symmetry breaking field to the boundary, $\delta H = -\sum_{i \in \text{bound}} \vec{h}_1 \cdot \vec{S}_i$. It is believed that for $h_1 \neq 0$ the model (1.1) has a single phase transition at $K = K_c$ where the boundary realizes just one universality class—the normal class—for all values of $\kappa$. The values of $q(N)$ and $N_c$ in the extraordinary-log phase (with $h_1 = 0$) are then determined by certain universal boundary OPE (operator product expansion) coefficients $\mu_\sigma$ and $\mu_t$ of the normal universality class (see section 2.2). These OPE coefficients can be extracted from the two-point function of the order parameter.

We would like to note that besides its relevance to the model with $h_1 = 0$, the normal universality class is interesting in its own right. In many ways, it is a natural target for conformal bootstrap due to the existence of two protected boundary operators (the tilt operator and the displacement operator discussed in section 2.2) and the sparseness of low-lying boundary operator spectrum anticipated from $e.g.$ the large-$N$ and $4 - \epsilon$ expansions. We analyze the bootstrap equations for the two-point function using two methods, which we call the truncated bootstrap and the positive bootstrap. The truncated bootstrap utilizes the method proposed by Gliozzi [39], where the operator spectrum is truncated by hand at a small number of low-lying operators. Our work here is a direct generalization of the application of the Gliozzi method to the normal boundary universality class of the 3d Ising

---

[4]Note that $N_c$ is almost certainly not an integer.

model in Ref. [6]. We find the OPE coefficients $\mu_\sigma$ and $\mu_t$ and hence the exponent $q$, see table 3 and figures 4a, 4b. This method gives $N_c$ just above 5, but unfortunately, the systematic errors associated with the truncation of the operator spectrum are difficult to estimate. The second method we employ, the positive bootstrap, does not suffer from such errors; however, it makes a crucial assumption of positivity of the coefficients in the expansion of the two-point function in the bulk channel. While we cannot prove that this assumption is valid, it is consistent with the $2 + \epsilon$, large-$N$ and $4 - \epsilon$ expansions of the two-point function (see Appendix C). It is also consistent with the results of the truncated bootstrap. Previously, the same assumption was made for the normal universality class of the Ising model, which was studied with the positive boostrap in Ref. [5]. Another assumption that we make is that the tilt and displacement operators are the lightest operators in the boundary channel—again, this assumption is consistent with the $2 + \epsilon$, large-$N$ and $4 - \epsilon$ expansions. Under the two assumptions above our positive bootstrap calculations produce rigorous bounds on $\mu_\sigma$ and $\mu_t$ and on $q$, allowing us to conclude $N_c > 3$, see figures 7, 8. In addition, we provide two qualitatively different sets of assumptions which imply that $N_c > 4$. One of these additional constraints is a stricter interval for $\mu_\sigma$, which is further verified by our truncated bootstrap results. We point out that our study here uses a more modern and powerful version of the positive bootstrap than Ref. [5], based on semi-definite programming techniques [40], and is, in fact, the first application of these techniques to boundary criticality.

We compare our bootstrap results to recent Monte Carlo studies of the extraordinary-log universality class [35, 37] and the forthcoming study of the normal universality class [41] in models with $N = 2, 3$, summarized in table 2. The Monte Carlo results are well within the bounds placed by positive bootstrap and in reasonable agreement with the truncated bootstrap results.

This paper is organized as follows. In section 2 we review the general bootstrap equations for a two-point function of a scalar field in the presence of a boundary and then discuss the particular form of these equations for the normal boundary universality class. Here we also discuss the existence of two protected boundary operators: the tilt operator and the displacement operator, as well as Ward identities associated with them. Section 3 reviews the framework of Ref. [31] for studying the boundary behavior of the $O(N)$ model in the $\kappa \gg 1$ region, explaining how the physics of the normal universality class is related to the model with no symmetry breaking field, $h_1 = 0$. Note that we present a derivation of the bulk + boundary action that differs slightly from that in the original paper [31] and illuminates why the particular ratio of OPE coefficients $\mu_\sigma/\mu_t$ enters the action. Section 4 analyzes the two-point function for the normal transition with the truncated bootstrap: results of this method for $\mu_\sigma$, $\mu_t$ and $q$ for various values of $N$ are presented here together with an attempt to estimate the systematic error associated with this method. Section 5 describes how the positive bootstrap can be used to place rigorous bounds on $\mu_\sigma$ and $\mu_t$. The results of positive bootstrap are presented in section 6 and compared to those of the truncated bootstrap and of the $2 + \epsilon$, large-$N$ and $4 - \epsilon$ expansions. Some concluding remarks are given in section 7. Various technical details are relegated to appendices. Appendix A gives a derivation of a Ward identity relating the ratio $\mu_\sigma/\mu_t$ to yet another universal coefficient associated with the boundary OPE of the $O(N)$ current. Appendix B presents some additional details and outputs of the positive bootstrap implementation. Appendix C collects the results for the two-point function for the normal universality class in $2 + \epsilon$, large-$N$ [31, 42] and $4 - \epsilon$ [13] expansions. We note that to our knowledge the $2 + \epsilon$ expansion results are new. Furthermore, while the $4 - \epsilon$ computation of the two-point function was performed in [13], we address a mixing problem that was not solved in this reference, and point out that the order $\epsilon$ results for the normal fixed point can be used to predict OPE data of the bulk CFT to higher loop order.

## 2 The boundary bootstrap

The concept of the conformal bootstrap dates back to the seventies [43, 44]. While it achieved its first remarkable successes in the realm of two-dimensional CFTs in the eighties [45], only in 2008 did it gain traction in the treatment of higher dimensional theories [46]. This breakthrough was made possible by the development of a numerical method that extracts rigorous bounds from the crossing equations without actually solving them, and it can be applied to systems obeying some positivity conditions which we will review shortly. We will refer to this approach as the *positive bootstrap*.

A large and ever growing literature has sprouted in the years since then [47]. The numerical approach was later complemented by analytic techniques, which will not be the focus of this work. More to the point, a different technology was proposed in [39] by Gliozzi. It allows to extract information from the crossing equation even when positivity is not guaranteed. This broader applicability is especially important for us because our setup falls in this category. However, Gliozzi's method, which we refer to as the *truncated bootstrap*, generates solutions affected by a systematic error that is not easy to estimate.

The positive bootstrap was first applied to CFTs with a boundary in [5], while the truncated bootstrap was employed for the same class of systems in [6]. In the present work, we use both methods. In the next subsection, we briefly review the basics of the conformal bootstrap in its BCFT incarnation. A more comprehensive account of the universal features of defects in CFTs can be found in [7, 48]. In subsection 2.2, we then state the specific problem considered in this work.

### 2.1 The crossing equation for the two-point function

Consider a $d$-dimensional CFT with a codimension-1 boundary – say, at $x^d = 0$. Our main focus is the two-point function of identical real scalar primary[5] operators placed away from the boundary:

$$\langle \phi(x)\phi(y) \rangle. \tag{2.1}$$

The full conformal symmetry of the theory is broken down to a subgroup that leaves $x^d = 0$ invariant. The following cross ratio is invariant under this subgroup:

$$\xi = \frac{(x-y)^2}{4x^d y^d}. \tag{2.2}$$

The existence of an invariant quantity composed of two points $x = (\mathbf{x}, x^d)$ and $y = (\mathbf{y}, y^d)$ implies that the two-point functions can be fixed by symmetry only up to a function of $\xi$.

Crucially, the correlation function (2.1) admits two operator product expansions (OPEs) with overlapping regions of convergence. The first OPE channel, the bulk channel, is the usual fusion of the two operators:

$$\phi(x) \times \phi(y) = \frac{1}{(x-y)^{2\Delta_\phi}} + \sum_k c_k \, C_k[x-y, \partial_y]\mathcal{O}_k(y), \qquad \text{bulk OPE.} \tag{2.3}$$

Here $\Delta_\phi$ is the scaling dimension of the external operators, the sum runs over all even spin primaries of the theory except the identity, $c_k$ are real OPE coefficients in a reflection positive theory, and the differential operators $C_k[x-y, \partial_y]$ are fixed by the $SO(d+1,1)$ conformal symmetry [43]. We suppressed the Lorentz indices of the operators $\mathcal{O}_k$ to avoid clutter, and in fact, as we shall see in a

---

[5]We define primary operators in the higher dimensional sense: these are the operators which are left invariant by the special conformal transformations which fix their insertion point.

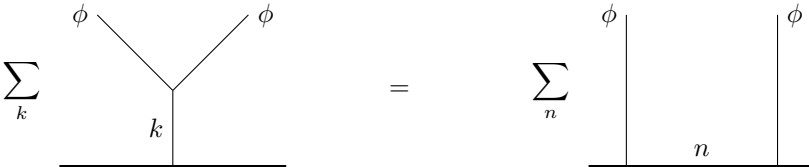

**Figure 3**: Depiction of crossing symmetry for the two-point function in BCFT.

moment, only the scalar primaries are relevant in this work. The second OPE channel, the boundary channel, is specific to BCFTs, and consists of replacing a local bulk operator with a sum of operators on the boundary, which we denote with a hat:

$$\phi(x) = \frac{a_\phi}{(2x^d)^{\Delta_\phi}} + \sum_n b_n \, \widehat{C}_n \big[x^d, \partial_{\mathbf{x}}^2\big] \, \widehat{\mathcal{O}}_n(\mathbf{x}), \qquad \text{boundary OPE.} \tag{2.4}$$

Again, we singled out the identity among the defect primary operators and distinguished its OPE coefficient with the letter $a$. The other boundary OPE coefficients, denoted as $b_n$, are again real as long as the BCFT satisfies reflection positivity, and the differential operators $\widehat{C}_n$ are determined by the $SO(d, 1)$ symmetry of the setup. The sum runs over boundary scalar primaries.

After using eq. (2.3) and eq. (2.4) separately to compute the two-point function eq. (2.1), one finds the crossing equation [48]

$$1 + \sum_k \lambda_k f_{\text{bulk}}(\Delta_k, \xi) = \xi^{\Delta_\phi} \left( \mu_\phi + \sum_n \mu_n f_{\text{bry}}(\widehat{\Delta}_n, \xi) \right). \tag{2.5}$$

This equation, illustrated in figure 3, is the starting point of the bootstrap, so let us describe in detail its ingredients. The left-hand side is, up to a kinematical factor, the expectation value of the bulk OPE (2.3). Of the primaries on the right hand side of (2.3), only the scalar ones acquire nonzero expectation values [4], which are fixed by symmetry up to proportionality constants $a_k$. Correspondingly, we defined $\lambda_k = a_k c_k$. Acting with the differential operators $C_k$ on the one-point functions, one obtains the conformal blocks $f_{\text{bulk}}(\Delta_k, \xi)$, where $\Delta_k$ is the scaling dimension of $\mathcal{O}_k$. As for the right hand side of eq. (2.5), *i.e.* the boundary channel, the sum runs over the same boundary operators as in eq. (2.4), and again the conformal blocks $f_{\text{bry}}(\widehat{\Delta}_n, \xi)$ are the avatars of the differential operators $\widehat{C}_n$. They depend on $\widehat{\Delta}_n$, the dimensions of the $\widehat{\mathcal{O}}_n$'s. Since each of the external operators $\phi$ is separately fused with the boundary, the coefficient of each conformal block is $\mu_n = b_n^2$, except for $\mu_\phi = a_\phi^2$.

The conformal blocks are known in closed form [48][6]:

$$f_{\text{bulk}}(\Delta, \xi) = \xi^{\Delta/2} \, _2F_1\left(\frac{\Delta}{2}, \frac{\Delta}{2}; \Delta + 1 - \frac{d}{2}, -\xi\right), \tag{2.6}$$

$$f_{\text{bry}}(\Delta, \xi) = \xi^{-\Delta} \, _2F_1\left(\Delta, \Delta + 1 - \frac{d}{2}; 2\Delta + 2 - d; -\frac{1}{\xi}\right). \tag{2.7}$$

For the moment, we emphasize a simple property of these functions. The bulk channel blocks admit a power series expansion around $\xi = 0$, while the boundary channel blocks can be expanded in powers of

---

[6]Our normalization for the leading contribution of a conformal family to the boundary OPE is

$$\phi(x) \sim (2x^d)^{-\Delta_\phi + \widehat{\Delta}_n} b_n \widehat{\mathcal{O}}_n(\mathbf{x}).$$

$1/\xi$. This is a simple consequence of scale invariance applied to the OPEs (2.3) and (2.4). While the contribution of heavy operators is suppressed in the bulk OPE at small $\xi$, heavy boundary operators are suppressed at large $\xi$. Hence, crossing equates two quite different representations of the same function.

Since the sums on both sides of eq. (2.5) are generically infinite, it is not obvious how to extract concrete information from it. This is the problem addressed by the truncated and the positive bootstrap, described in detail in sections 4 and 5 respectively. For the moment, let us move on to the $O(N)$ model, and describe what is known about eq. (2.5) in that case.

## 2.2 The boundary bootstrap for the normal transition

As described in the introduction, the $O(N)$ model in $d = 3$ is believed to admit a normal conformal boundary condition which breaks $O(N)$ to $O(N - 1)$. Our focus is the two-point function of the lowest dimensional primary in the vector representation of $O(N)$ in the presence of this boundary condition. One can think of this operator as the continuum limit of the lattice spin which appears in eq. (1.1). Clearly, the bulk channel OPE is unaffected by the boundary, and it can be organized in representations of $O(N)$. On the other hand, boundary operators carry $O(N - 1)$ indices.

To fix conventions, we denote by letters like $a$, $b$ the indices in the fundamental representation of $O(N)$, and we split them as $a = (i, N)$, where indices like $i$, $j = 1, \ldots, N - 1$ run over the subgroup unbroken by the boundary magnetic field. With these conventions, the lightest $O(N)$ vector $\phi_a$ has the following fusion rule:

$$\phi_a \times \phi_b \sim \sum_S \delta_{ab} \mathcal{O} + \sum_T \mathcal{O}_{(ab)} + \sum_A \mathcal{O}_{[ab]}, \tag{2.8}$$

where the $O(N)$ singlets $S$ and the traceless symmetric tensors $T$ have even $O(d)$ spin, while the antisymmetric tensors $A$ have odd $O(d)$ spin. Since only scalar primaries acquire one-point functions in the presence of a boundary, only the singlet and the traceless symmetric representations survive in the bulk channel decomposition of the two-point function.

For the boundary channel, we discuss separately the *longitudinal* component $\phi_N \equiv \sigma$. As an $O(N - 1)$ singlet, $\sigma$ can acquire an expectation value, hence its boundary fusion rule reads

$$\sigma \sim 1 + \sum_{\widehat{S}} \widehat{\mathcal{O}}, \tag{2.9}$$

where $\widehat{S}$ denotes scalar boundary primaries of non-vanishing dimensions, which are also singlets under $O(N - 1)$. On the other hand, the *transverse* components $\phi_i$ only admit $O(N - 1)$ vectors in their boundary OPE, which we denote as $\widehat{V}$:

$$\phi_i \sim \sum_{\widehat{V}} \widehat{\mathcal{O}}_i. \tag{2.10}$$

Let us look in more detail at the low lying spectrum, starting with the bulk channel. All the $O(N)$ models have one relevant operator in the singlet scalar sector, $\epsilon$, which couples to the temperature. The leading operators in the vector ($\phi$) and symmetric traceless ($T$) representations are also relevant. Precise estimates are available for both the dimensions of these operators and the dimension $\Delta_{\epsilon'}$ of the least irrelevant operator responsible for the leading correction to scaling. We report in table 1 the values we use as input in this work.[7]

---

[7]After this work was completed, Ref. [49] appeared with improved Monte Carlo estimates of critical exponents for $N = 4, 5$, and 10. We checked that using these new estimates did not change any of the main conclusions of the paper.

| $N$ | $\Delta_\phi$ | $\Delta_\epsilon$ | $\Delta_{\epsilon'}$ | $\Delta_T$ | Method [Ref.] |
|-----|---------------|-------------------|----------------------|------------|---------------|
| 1 | 0.518149(**10**) | 1.412625(**10**) | 3.82968(23) | - | CB [50][51] |
| 2 | 0.519088(**22**) | 1.51136(**22**) | 3.794(8) | 1.23629(**11**) | CB [52] |
| 3 | 0.518936(**67**) | 1.59488(**81**) | 3.759(2) | 1.20954(**32**) | CB[53], MC [54] |
| 4 | 0.5190(**15**) | 1.660(**15**) | 3.765 | 1.1864(34) | CB[55][56], MC [57] |
| 5 | 0.51690(55) | 1.7174(15) | 3.760(18) | 1.1568(10) | DE[58], CB[56] |
| 10 | 0.51155(30) | 1.8605(13) | 3.807(7) | 1.1003(10) | DE[58], CB[56] |
| 20 | 0.50645(15) | 1.93719(68) | 3.887(2) | 1.0687(10) | DE[58], CB[56] |
| $\infty$ | 1/2 | 2 | 4 | 1 | |

**Table 1**: Bulk critical exponents from the literature. CB stands for conformal bootstrap, MC for Monte Carlo and DE for derivative expansion. References [51], [54] and [57] are only used for $\Delta_{\epsilon'}$. In particular, the value from [57] was taken from its table 10, where the error is not reported. Reference [56] is only used for $\Delta_T$, and the remaining assignments are unambiguous. As for the errors, the ones in the CB come in two flavors. Values in bold indicate rigorous errors, while values in regular font refer to the single correlator bootstrap, where the uncertainty is not rigorous.

Let us turn to the boundary channel. At the normal fixed point, there are two protected boundary operators, one in the singlet $\widehat{S}$ channel and one in the vector $\widehat{V}$ channel. The former is the *displacement operator* D, with dimension $\widehat{\Delta}_D = 3$. Its existence on any conformal defect is guaranteed [7, 59], and a Ward identity enforces its presence in the boundary OPE of all bulk operators with a one-point function. Specifically,

$$\sigma(x) \sim \frac{a_\sigma}{(2x^3)^{\Delta_\phi}} + b_D(2x^3)^{3-\Delta_\phi}D(\mathbf{x}), \qquad x^3 \to 0, \qquad \mu_\sigma = a_\sigma^2, \qquad \mu_D = b_D^2, \qquad (2.11)$$

and

$$\frac{\mu_\sigma}{\mu_D} = \left(\frac{4\pi}{\Delta_\phi}\right)^2 C_D. \qquad (2.12)$$

In this formula, $C_D$ is a strictly positive constant in a reflection positive theory.[8] The protected operator in the vector channel, which we call the *tilt operator* $t_i$, has dimension $\widehat{\Delta}_t = 2$ [60], and is assumed to be the lightest operator in the boundary spectrum:

$$\phi_i(x) \sim b_t(2x^3)^{2-\Delta_\phi}t_i(\mathbf{x}), \quad x^3 \to 0 , \quad \mu_t = b_t^2. \qquad (2.13)$$

Just as the displacement operator couples to a deformation of the boundary, the tilt operator couples to a change in the direction of the boundary magnetic field. The same argument that leads to eq. (2.12) also provides a relation of three OPE coefficients in this case:

$$\frac{\mu_\sigma}{\mu_t} = 16\pi^2 C_t. \qquad (2.14)$$

---

[8]The displacement also appears as the leading term in the boundary OPE of the stress tensor: $T^{33} \sim -\sqrt{C_D}D$. Equivalently, this equation can be written as a Ward identity: $\partial_\mu T^{\mu 3} = -\delta(x^3)\sqrt{C_D}D$. Then, $\sqrt{C_D} \neq 0$ because translational invariance is broken by the boundary, and furthermore $\sqrt{C_D} \in \mathbb{R}$ by reflection positivity as for any other OPE coefficient. All operators in this paper are normalized to one, but in the natural normalization $D \to D/\sqrt{C_D}$, so that $C_D$ is often defined as the squared norm of the displacement in radial quantization.

Here, $C_{\mathrm{t}}$ determines the coefficient of the tilt operator in the boundary OPE of the $\mathrm{O}(N)$ symmetry current $j^{\mu}_{[ab]}$, where $a$ and $b$ are anti-symmetrized:

$$j^{3}_{[Ni]}(x) \sim \sqrt{C_{\mathrm{t}}}\, \mathrm{t}_i(\mathbf{x}), \qquad x^3 \to 0. \tag{2.15}$$

In fact, the tilt operator is the leading operator in that OPE, and the only scalar. Its contribution is nicely captured by the equation

$$\partial_\mu j^\mu_{[Ni]} = \delta(x^3)\sqrt{C_{\mathrm{t}}}\, \mathrm{t}_i. \tag{2.16}$$

Eq. (2.16) has an obvious generalization to any defect, while eq. (2.15) shows that, in the codimension one case, $\mathrm{t}_i$ is identified with the boundary value of the appropriate component of the current. For completeness, we review the derivation of eq. (2.14) in appendix A. Because it determines the fate of the extraordinary-log phase, as we review in section 3, the coefficient $C_{\mathrm{t}}$ is the main target of this paper.

As for the rest of the boundary spectrum, both the large $N$ [42, 60] and the $4 - \epsilon$ expansions [13] indicate that there are no operators lighter than the displacement except for the tilt operator. In fact, the boundary spins are frozen at the normal transition, and the lattice intuition suggests that the only simple operators are built geometrically: locally deforming the position of the boundary and locally changing the orientation of the boundary spins. It is tempting to conjecture that the normal transition defines the conformal boundary condition with the largest possible gap above the protected operators. It would be interesting to test this possibility with the conformal bootstrap. In this work, we assume a weaker form of the conjecture, namely that the spectrum of boundary primary operators is as follows:

$$\widehat{\Delta} = \begin{cases} 2, & \text{tilt operator } \mathrm{t}_i \\ 3, & \text{displacement operator D} \\ \geq 3. \end{cases} \tag{2.17}$$

With this information about the spectrum, let us move on to the crossing equation, as defined by eq. (2.5). The two-point function of $\phi_a$ contains two $\mathrm{O}(N-1)$ singlets:

$$G_\sigma(x, y) = \langle \sigma(x)\sigma(y)\rangle\,, \tag{2.18}$$

$$G_\phi(x, y) = \frac{1}{N-1}\sum_i \langle \phi_i(x)\phi_i(y)\rangle\,. \tag{2.19}$$

Hence, there are two non trivial crossing equations, respectively:

$$1 + \sum_{k \in S} \lambda_k f_{\mathrm{bulk}}(\Delta_k, \xi) + \sum_{l \in T} \lambda_l f_{\mathrm{bulk}}(\Delta_l, \xi) = \xi^{\Delta_\phi}\left(\mu_\sigma + \mu_D f_{\mathrm{bry}}(3, \xi) + \sum_{\substack{n \in \widehat{S} \\ \widehat{\Delta}_n \geq 3}} \mu_n f_{\mathrm{bry}}(\widehat{\Delta}_n, \xi)\right), \tag{2.20}$$

and

$$1 + \sum_{k \in S} \lambda_k f_{\mathrm{bulk}}(\Delta_k, \xi) - \frac{1}{N-1}\sum_{l \in T} \lambda_l f_{\mathrm{bulk}}(\Delta_l, \xi) = \xi^{\Delta_\phi}\left(\mu_{\mathrm{t}} f_{\mathrm{bry}}(2, \xi) + \sum_{\substack{n \in \widehat{V} \\ \widehat{\Delta}_n \geq 3}} \mu_n f_{\mathrm{bry}}(\widehat{\Delta}_n, \xi)\right). \tag{2.21}$$

Our focus is the linear combination $G_S = G_\sigma + (N-1)G_\phi$, which projects the bulk channel onto the $O(N)$ singlet:

$$N + \sum_{k \in S} N\lambda_k f_{\text{bulk}}(\Delta_k, \xi) = \xi^{\Delta_\phi}\left(\mu_\sigma + (N-1)\mu_{\text{t}}f_{\text{bry}}(2, \xi) + \mu_{\text{D}}f_{\text{bry}}(3, \xi) + \sum_{\widehat{\Delta}_n \geq 3} \tilde{\mu}_n f_{\text{bry}}(\widehat{\Delta}_n, \xi)\right).$$
$$(2.22)$$

In the boundary channel, we collected together all operators with dimensions equal or above that of the displacement. Correspondingly, $\tilde{\mu}_n = \mu_n$ for $\widehat{S}$ operators, and $\tilde{\mu}_n = (N-1)\mu_n$ for $\widehat{V}$ operators. Of course, other linear combinations contain information about the traceless symmetric spectrum in the bulk. In particular, $G_\sigma - G_\phi$ projects out the singlet. We will not explore the related constraint in this paper.

## 3   Intermezzo: from the normal fixed point to the extraordinary phase

We now review the main idea of [31], which allows us to relate the OPE data of the normal transition to the scenarios for the boundary phase diagram of the $O(N)$ model without any explicit symmetry breaking field, as we described them in the introduction. Our presentation here differs slightly from that in [31], elucidating why the ratio of OPE coefficients $\mu_\sigma/\mu_t$ plays a prominent role.

We will be interested in a model where at some intermediate length-scale (much larger than the UV cut-off) the boundary spontaneously breaks the $O(N)$ symmetry. At this length scale, we expect the system to flow to the normal fixed point. This is true of the lattice model (1.1) in the regime $\kappa \gg 1$. Indeed, when $\kappa = \infty$, the boundary spins are frozen along some fixed direction, acting as a symmetry breaking field. When $\kappa$ is large but finite, the boundary order is expected to persist at least up to some large intermediate length scale. To describe the system on at distances equal to or larger than this length scale, we can start with the normal boundary fixed point and deform it with a perturbation that restores the full $O(N)$ symmetry at the level of the action.

One can restore the $O(N)$ symmetry by adding dynamical degrees of freedom which compensate for the variation of the original non-symmetric action. Let $S_{\text{normal}}$ be the bulk+boundary action at the normal fixed point, with the symmetry breaking field pointing along the $N$th direction. The variation of this action under the broken $O(N)$ rotation is captured by integrating the divergence of the current in the path integral:

$$\delta S_{\text{normal}} = \int d^3x\, \omega^i\, \partial_\mu j^\mu_{[Ni]}(x),\tag{3.1}$$

where $\omega^i$ are the infinitesimal angles which parametrize the rotation. We can then use the Ward identity (2.16) to write

$$\delta S_{\text{normal}} = \sqrt{C_{\text{t}}} \int d^2\mathbf{x}\, \omega^i\, \text{t}_i(\mathbf{x}).\tag{3.2}$$

We see that we can cancel this variation by introducing a coupling of the tilt operator with new boundary fields $\pi^i(\mathbf{x})$ and assign to them the transformation law

$$\delta\pi^i(\mathbf{x}) = -\omega^i.\tag{3.3}$$

The action

$$S_{\text{normal}} + \sqrt{C_{\text{t}}} \int d^2\mathbf{x}\, \pi^i(\mathbf{x})\, \text{t}_i(\mathbf{x})\tag{3.4}$$

is invariant under $O(N)$ at zeroth order in $\pi^i$. The final step is to make the new fields dynamical, and this can be achieved with a kinetic term invariant under the shift (3.3) and under the unbroken

$O(N-1)$ subgroup, which is realized linearly. It is natural to reinterpret the $\pi^i$'s as components of a $O(N)$ unit vector $\vec{n} = (\pi^i, \sqrt{1-\pi^2})$, and to add the non-linear sigma model to complete the action:

$$S = S_{\text{normal}} + \frac{1}{2g} \int d^2\mathbf{x} \, (\partial\vec{n}(\mathbf{x}))^2 + \sqrt{C_{\text{t}}} \int d^2\mathbf{x} \, \pi^i(\mathbf{x}) \, \mathbf{t}_i(\mathbf{x}). \tag{3.5}$$

When $g = 0$, the $\pi^i$ are frozen to be constants, and (3.5) reduces to the action of the normal fixed point, where the direction of the magnetisation is fixed by $\pi^i$. We will be interested in the stability of the $g = 0$ fixed point.

Because the coefficient of the last term in eq. (3.5) is fixed by the requirement of $O(N)$ invariance, it is not renormalized along the RG flow. On the contrary, the coupling $g$ has an interesting $\beta$ function, which we address shortly. The action (3.5) needs additional couplings to restore $O(N)$ invariance to higher orders in $\pi^i$: the coefficients of the ones which are relevant or marginal at $g = 0$ must be fine tuned, and their values do not affect $\beta(g)$ at quadratic order [31]. Finally, we check that there is no additional $O(N)$ singlet which is relevant or marginal at the normal fixed point. This fact follows from the spectra of the non-linear sigma model and of the normal fixed point. On one hand, with the exception of $(\partial n)^2$, there are no relevant or marginal operators built out of $\pi^i$ in two dimensions which are classially $O(N)$ invariant. On the other hand, the normal fixed point has no relevant $O(N-1)$ singlet operators at all, as we explained in section 2.2. The only $O(N-1)$-invariant marginal operators involving this sector are of the kind $f(\vec{\pi}^2)\pi^i\mathbf{t}_i$, but these are not classically $O(N)$ invariant either, and so they are exactly the operators whose coefficients will need to be fine-tuned order by order in $\pi^i$.

We conclude that there is a one-parameter flow controlled by the $\beta$ function of $g$, which reads [31]

$$\beta(g) = \alpha g^2 + O(g^3), \tag{3.6}$$

with

$$\alpha = \frac{\pi}{2} C_{\text{t}} - \frac{N-2}{2\pi} = \frac{1}{32\pi} \frac{\mu_\sigma}{\mu_{\text{t}}} - \frac{N-2}{2\pi}, \tag{3.7}$$

where we used eq. (2.14) in eq. (3.7).

Hence, the stability of the $g = 0$ fixed point is decided by the sign of $\alpha$. When $\alpha > 0$, a small initial $g$ flows to zero logarithmically and the extraordinary-log universality class is realized. Here the $O(N)$ invariant bulk correlation functions in the isotropic scaling limit $(\vec{x}, x^d) \to \lambda(\vec{x}, x^d)$, $\lambda \to \infty$, match those of the normal fixed point up to corrections in powers of $1/\log\lambda$ (the latter can be computed perturbatively in $g$). However, due to the logarithmic approach to the $g = 0$ fixed point and anomalous dimension of the field $\vec{n}$, correlation functions along the boundary decay logarithmically [31]:

$$\langle n^a(\vec{x})n^b(0) \rangle \sim \frac{\delta^{ab}}{(\log|\vec{x}|)^q}, \qquad q = \frac{N-1}{2\pi\alpha}. \tag{3.8}$$

We expect the two-point function of the bulk field $\langle \phi^a(x)\phi^b(y) \rangle$ to exhibit the same logarithmic decay in the limit $x^d$ fixed, $|\vec{x}| \to \infty$.

When $\alpha < 0$, the $g = 0$ fixed point is unstable, the long distance physics depends on the closest stable fixed point and various scenarios open up [31]. We will not attempt to resolve the infra-red physics in this regime here, instead, we concentrate on computing the value of $\alpha$.

Equation (3.7) is a concrete target for the bootstrap. We know that $\alpha > 0$ for $N = 2$ because $C_{\text{t}} > 0$, and vice versa $\alpha < 0$ at large $N$ [31, 42, 60]. In the following sections, we use the truncated and the positive bootstrap to identify the sign and magnitude of $\alpha$ in part of the remaining range. Our findings are consistent with $\alpha$ changing sign once for $N > 2$; we let $N = N_c > 2$ be the zero of $\alpha$,

| $N$ | $\mu_\sigma$ | $\mu_{\rm t}$ | $\alpha_{\rm norm}$ | $\alpha_{\rm eo}$ |
|------|------|------|------|------|
| 2 [37, 41] | 8.29(1) | 0.276(4) | 0.300(5) | 0.27(1) |
| 3 [35, 41] | 9.83(1) | 0.280(3) | 0.190(4) | 0.15(2) |

**Table 2**: Monte Carlo results for surface criticality in the classical O($N$) model. The values of $\mu_\sigma$ and $\mu_{\rm t}$ are from the forthcoming study [41] of the normal boundary universality class, and $\alpha_{\rm norm}$ is obtained from these using (3.7). $\alpha_{\rm eo}$ is from the studies of the extraordinary phase (with $h_1 = 0$), where it was extracted from the boundary correlation function (3.8) [35, 37].

such that the extraordinary-log universality class is realized for $2 \leq N < N_c$. Extracting information about $N_c$ is a key goal of this work.

Recent Monte Carlo simulations of the O($N$) model support the existence of the extraordinary-log universality class for $N = 2$ and $N = 3$ [35, 37]. The value of $\alpha$ in these models has been extracted from the logarithmic decay of the boundary two-point function, eq. (3.8), and is listed in table 2 as $\alpha_{\rm eo}$. A forthcoming Monte Carlo study [41] directly investigates the normal universality class in models with $N = 2, 3$ and extracts the values of $\mu_\sigma$, $\mu_{\rm t}$. We collect these data in table 2 together with the associated value of $\alpha$ obtained via eq. (3.7) and listed as $\alpha_{\rm norm}$. We note that $\alpha_{\rm eo}$ and $\alpha_{\rm norm}$ found by Monte Carlo are in reasonable agreement, as predicted by the RG analysis above. Since $\alpha_{\rm norm}$ has slightly smaller error bars, to simplify the presentation we will use $\alpha_{\rm norm}$ when comparing our bootstrap results for $\alpha$ to Monte Carlo. In fact, as pointed out in [35], the error bar on $\alpha_{\rm eo}$ needs to be taken with a grain of salt, given the difficulty of fitting the function (1.2) and the presence of slowly decaying subleading corrections that are not accounted for in the fit.

## 4 The truncated bootstrap

The gist of the truncation method [39] is that finite truncations of the infinite sums in the crossing equation (2.5) provide an approximation to the low lying CFT data. In the boundary bootstrap, given the lack of rigorous positivity constraints for the bulk channel OPE coefficients, the truncation method proves to be a natural starting point for exploration. It makes no assumption of positivity and has been used to explore both unitary and non-unitary CFTs [6, 61–65]. However, truncating the OPE is plagued by systematic errors that in most cases are difficult to estimate, as we expound later in this section.

Consider the crossing equation (2.22), which involves both OPE coefficients of interest to us, $\mu_\sigma$ and $\mu_{\rm t}$. To start with, we linearize the constraint by expanding around a value of the cross-ratio $\xi$ where the regions of convergence of both channels (i.e., bulk and boundary) overlap. In the boundary bootstrap literature, this is usually chosen to be $\xi = 1$. Simultaneously, we also truncate the infinite sums to a finite number of operators. We label the truncations of eq. (2.22) in this paper by pairs of integers $(n_{\rm bulk}, n_{\rm bry})$ standing for the number of operators left in each channel (not counting for the bulk identity which is always present). The linear system of equations hence obtained from the $G_S$ crossing equation (2.22) look like

$$-\sum_{k=1}^{n_{\rm bulk}} N\lambda_k f_{\rm bulk}(\Delta_k, 1) + \mu_\sigma + (N-1)\mu_{\rm t} f_{\rm bry}(2,1) + \mu_D f_{\rm bry}(3,1) + \sum_{n=1}^{n_{\rm bry}-3} \tilde{\mu}_n f_{\rm bry}(\widehat{\Delta}_n, 1) = N, \quad (4.1)$$

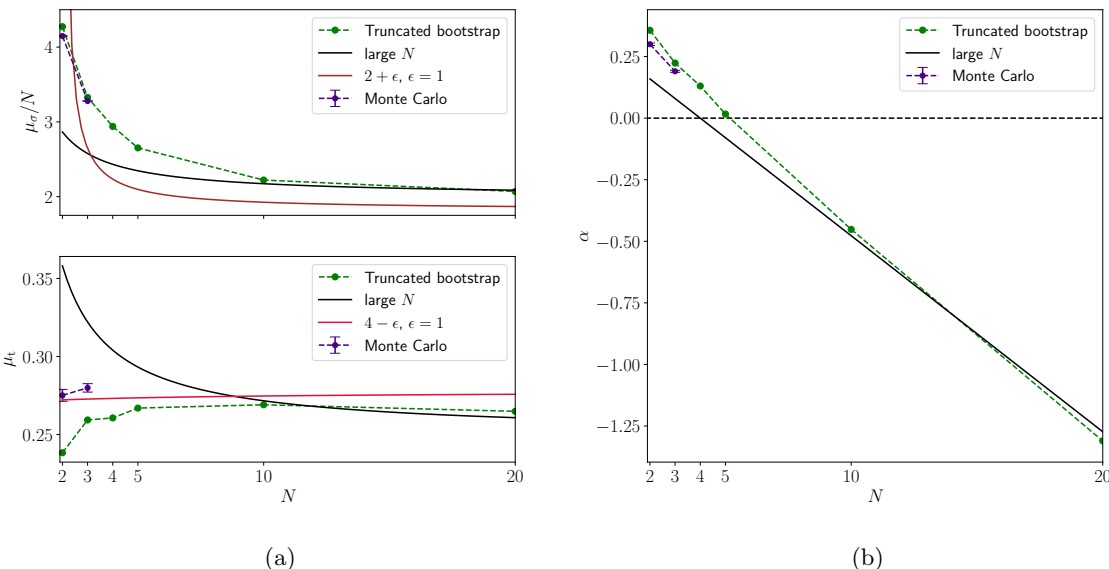

**Figure 4**: OPE coefficients from the truncated bootstrap. $\mu_\sigma$ and $\mu_\mathrm{t}$ estimates (on the left) compared to selected perturbative results (Appendix C) and Monte Carlo results in [41]. Corresponding values of $\alpha$ are on the right. The value of $N_c$ is determined by $\alpha(N_c) = 0$.

$$-\left(\sum_{k=1}^{n_\mathrm{bulk}} N\lambda_k \, \partial_\xi^m f_\mathrm{bulk}(\Delta_k,\xi)\big|_{\xi=1}\right) + (\Delta_\phi)_m \mu_\sigma + (N-1)\mu_\mathrm{t} \, \partial_\xi^m(\xi^{\Delta_\phi} f_\mathrm{bry}(2,\xi))\big|_{\xi=1}$$

$$+ \mu_D \, \partial_\xi^m(\xi^{\Delta_\phi} f_\mathrm{bry}(3,\xi))\big|_{\xi=1} + \sum_{n=1}^{n_\mathrm{bry}-3} \tilde{\mu}_n \, \partial_\xi^m(\xi^{\Delta_\phi} f_\mathrm{bry}(\widehat{\Delta}_n,\xi))\big|_{\xi=1} = 0, \quad (4.2)$$

where $(\Delta_\phi)_m$ is the Pochhammer symbol. Of course, there are infinitely many homogeneous equations of the form (4.2) labeled by the integer $m$, of which we keep only the first $M$. The homogeneous system of equations involving the derivatives of conformal blocks is linear in the vector of the OPE coefficients $(N\lambda_k, \mu_\sigma, (N-1)\mu_\mathrm{t}, \mu_D, \tilde{\mu}_n)$ of dimension $L = n_\mathrm{bulk} + n_\mathrm{bry}$. The system is over-constrained if we choose $M > L$, and has a non-trivial solution only if the smallest singular value of the matrix of derivatives is zero [62].

Practically, we use the known critical exponents to fix the scaling dimension of the lowest operators in the spectrum (table 1), which further reduces the dimensionality of the search space for the scaling dimensions of other operators. Notice in particular that the dimension $\Delta_\phi$ of the external operators is always an input in this section. Then we minimize the smallest singular value of the matrix corresponding to eq. (4.2) and hope to find approximate zeros. Once the unknown dimensions are found, we can solve the system of homogeneous equations along with the inhomogeneous equation to obtain the OPE coefficients. At this stage, we must also discard solutions that do not satisfy the unitarity constraints

$$\mu_i \geq 0 \qquad\qquad (4.3)$$

for all the boundary OPE coefficients.

| $N$ | $\Delta_3$ | $\Delta_4$ | $\mu_\sigma$ | $\mu_\mathrm{t}$ | $\alpha$ | $\mu_D \times 100$ |
|---|---|---|---|---|---|---|
| 2 | 7.007 | 12.489 | 8.546 | 0.2383 | 0.3567 | 7.298 |
| 3 | 6.883 | 12.385 | 9.98 | 0.2593 | 0.2236 | 7.236 |
| 4 | 6.845 | 12.358 | 11.758 | 0.2606 | 0.1304 | 7.609 |
| 5 | 6.819 | 12.351 | 13.259 | 0.2669 | 0.01660 | 7.038 |
| 10 | 6.845 | 12.414 | 22.220 | 0.2691 | -0.4518 | 5.512 |
| 20 | 6.955 | 12.550 | 41.386 | 0.2649 | -1.311 | 3.000 |

**Table 3**: CFT data from the $(4, 3)$ truncation for integer values of $N \geq 2$ with input data from table 1. $\Delta_3$, $\Delta_4$ refer to dimensions of unknown bulk operators that are found by minimization. Values are only reported up to first few significant digits.

The first truncation we consider is $(4, 3)$, with the scaling dimensions of the lowest two bulk operators fixed from the literature, *i.e.* $\Delta_\epsilon$ and $\Delta_{\epsilon'}$ in table 1. Each bulk/boundary operator in the truncation contributes two pieces of CFT data: the scaling dimension and the OPE coefficient. Of the 14 parameters obtained for the $(4, 3)$ truncation, the dimensions of the protected boundary operators (boundary identity, displacement and tilt) and two bulk operators are known *a priori*. Thus, this leaves us with $14 - 3 - 2 = 9$ independent parameters in the non-linear system. As there are $M + 1$ equations constraining the parameters, a choice of $M = 8$ may result in either isolated solutions or no solution at all [66]. It turns out that we do find good solutions for all finite values of $N$ considered in table 1. The solutions obtained from the canonical `FindMinimum` package in `Mathematica` appear to be true zeros of the smallest singular value $z$ of the derivative matrix. This is confirmed by changing the precision "`prec`" and observing that

$$\mathrm{Min}\,[\log z] \propto -\texttt{prec}.$$

In other words, increasing the precision at which the minimization is run simply makes the zeros more precise without altering the CFT data. It would be interesting to understand the origin of this fact, perhaps along the lines of [67]. The OPE coefficients obtained this way for integer values of $N \geq 2$ from table 1 are plotted in figure 4a. Figure 4b shows the corresponding values of $\alpha$ and indicates that the critical value $N_c$ estimated from the truncation $(4, 3)$ is $N_c \sim 5$. The full set of unknown CFT data obtained from the $(4, 3)$ truncation is tabulated in table 3. We used a precision of 200 for the minimization, but we only report here the first few significant digits of the CFT data.

Let us now focus on the integer values of $N \leq 5$. For $N = 1$ our analysis excellently corroborates the results obtained in Ref. [6].[9] Following the approach of [6], we test the stability of the solutions for $N = 2$ and $N = 3$ by adding an extra operator in either channel, that is, by considering the truncations $(5, 3)$ and $(4, 4)$. If we choose $M = 9$, there is one free parameter in the set of CFT data for both extended truncations $(16 - 3 - 2 - 10 = 1)$, and we should generically expect to find a one-parameter family (OPF) of solutions for each truncation. Indeed, we find these families as functions of the scaling dimensions of the added operators, $\Delta_5$ and $\widehat{\Delta}_4$. For example, the $(5, 3)$ family for $N = 2$ is sketched in Fig. 5. The truncation $(4, 3)$ corresponds to $\Delta_5 \to \infty$ and $\widehat{\Delta}_4 \to \infty$, which is in line

---

[9]For example, for the $(4, 2)$ truncation, our results for the two unknown dimensions are $\Delta_3 = 7.311$ and $\Delta_4 = 13.036$ compared to the quoted values of $\Delta_3 = 7.316(14)$ and $\Delta_4 = 13.05(4)$ in [6]. In the journal version of the same paper, data for the normal transition in the O(2) and O(3) models appear as well. However, the truncation used there was incomplete, since it did not include the tilt operator.

with the numerical solutions because all the CFT data approach the $(4, 3)$ solution monotonically as we increase the free parameter in either family.

Reference [6] used the OPFs to obtain an estimate of the systematic error due to the truncation: if the low lying CFT data depended weakly on the scaling dimension of the additional operator in an extended truncation, we might trust their values as obtained from the original truncation. For $N = 2$ we find that the $(5, 3)$ OPF exists for $\Delta_5 \geq 15.96(2)$. The end of this family corresponds to $\mu_t$ going to zero and eventually becoming negative. When $\mu_t$ is arbitrarily close to zero with $\mu_\sigma$ finite, $\alpha$ can be arbitrarily large, i.e. this method of estimating the error provides no approximate upper bound on $\alpha$. However, the $(4, 4)$ OPF exists for $\widehat{\Delta}_4 \geq 4.66(2)$ and provides an approximate *lower* bound on $\alpha$ at its end: $\alpha(2) \geq 0.144$. Figure 6 illustrates the change in the OPE coefficients along both these families. Note that our result for $\mu_t$ appears to vary more upon adding an extra operator than $\mu_\sigma$ does.

For $N = 3$ we obtain a similar picture, with the lower bound $\alpha(3) \geq -0.047$ coming from the $(4, 4)$ OPF. Based on this analysis, it seems that we cannot confidently place $N = 3$ above or below $N_c$. However, it is unclear if the use of the OPFs is a correct way to estimate systematic errors, which might be overestimated. An in-depth analysis of the truncation error is lacking in the literature and is beyond the scope of this work.

Some more information on the truncation error can be gathered by comparison with other methods. The CFT data obtained from the $(4, 3)$ truncation for all studied values of $N$ sit squarely in the middle of the positive bootstrap bounds in figures 7, 8. However, a more detailed analysis of the island of solutions to crossing shows that the truncated data actually lie slightly *outside* the allowed region, at least for $N = 3$—see figure 10. Hence, the truncation error cannot be too small: as we discuss below, we expect it to be larger on $\mu_t$ than on $\mu_\sigma$.

The $(4, 3)$ truncation for $N \gtrsim 10$ also agrees reasonably well with estimates from large $N$ calculations (figures 4a, 4b). Large $N$ expansion indicates that $\mu_t$ is a decreasing function of $N$ for $N \to \infty$, so curiously, combined with our truncated bootstrap findings, this would imply that $\mu_t$ is a non-monotonic function of $N$.

We now compare the results of the $(4, 3)$ truncation to Monte Carlo data on the normal transition for $N = 2, 3$ [41] listed in table 2. We find that our value of $\mu_\sigma$ is about 1-3% larger than the Monte Carlo result, while the value of $\mu_t$ is about 7-15% smaller. While both values are well outside the Monte Carlo error-bars, we consider the agreement to be very good, given the pessimistic estimates of possible truncation error discussed above. The larger deviation of our $\mu_t$ from Monte Carlo as compared to $\mu_\sigma$ might be due to stronger sensitivity of $\mu_t$ to the addition of extra boundary operators, as shown in Fig. 6, right. Overall, for $N = 2, 3$ the deviations in $\mu_\sigma$ and more importantly $\mu_t$ result in our value of $\alpha$ being somewhat larger than the Monte Carlo result. Note, however, that if we assume in our $N = 4$ results the same 15% error on $\mu_t$ and 3% error on $\mu_\sigma$, we still obtain $\alpha(4) \approx 0.06$, i.e. $N_c > 4$.

## 5   The positive semi-definite bootstrap

As we have shown, the truncated bootstrap, when applied to $O(N)$ vector models with a boundary, yields solutions that agree with asymptotics from large-$N$ expansion, but also have uncontrolled systematic errors that *can* be large in magnitude, as far as we have investigated. Therefore, we also approach the problem using the more mainstream positive bootstrap [68].

Unlike the truncated bootstrap, the positive bootstrap requires that the coefficients $\lambda_k$ in eq. (2.5) are positive. If this condition is verified, then the positive bootstrap yields rigorous bounds. However, the positivity of the bulk channel OPE coefficients cannot be guaranteed on general grounds. The

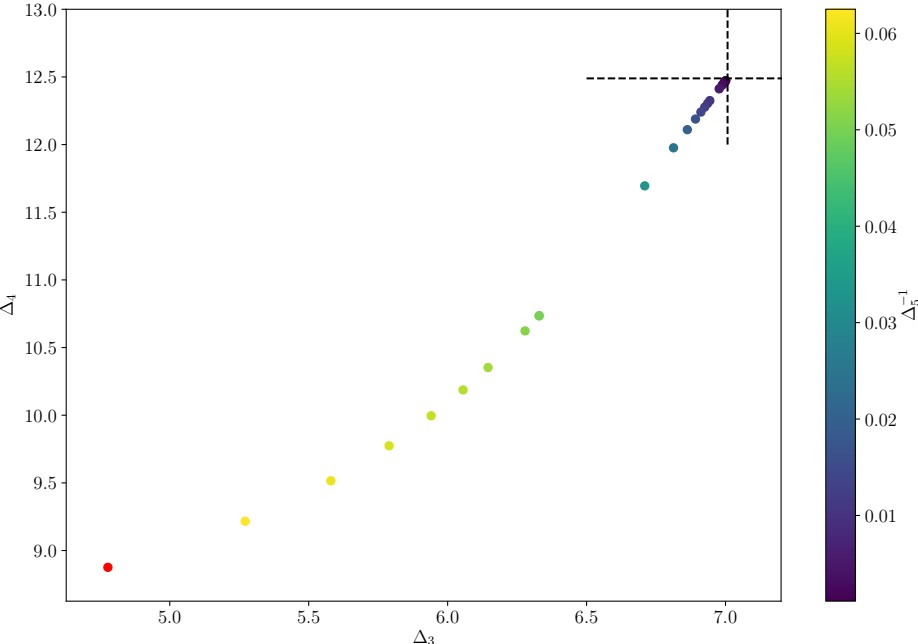

**Figure 5**: One-parameter family of solutions in the $(5,3)$ truncation for $N = 2$. We label the bulk channel operators as $(\Delta_\epsilon, \Delta_{\epsilon'}, \Delta_3, \Delta_4, \Delta_5)$ where the first two are fixed from the literature (table 1).The solution from the $(4,3)$ truncation is marked with a crosshair of dashed lines. The red point in the bottom left corner is a solution that has negative $\mu_t$, which is not allowed.

authors of [5] conjectured that any CFT allows for at least one boundary condition for which the bulk OPE coefficients are positive. They further assumed that this is true for both the special and the normal universality classes in the Ising model. This was qualified in [6] where more evidence was provided for the positivity to be associated with the normal transition. In this section, we must assume the stronger version of the conjecture that specifically, the normal transition is a boundary condition that manifests positivity in the bulk channel for any value of $N$. This is consistent with the results of $2 + \epsilon$, large-$N$ and $4 - \epsilon$ expansions presented in Appendix C. The assumption is also consistent with our results from truncated bootstrap, where none of the truncated solutions came with negative $\lambda_k$.

In this work, we use the latest semi-definite programming (SDP) package SDPB 2.0 [69] that is specifically tailored for the SDP problems one encounters in conformal bootstrap. Starting from the crossing equation (2.22) for $G_S$, positive bootstrap can be employed to place rigorous bounds on the OPE coefficients of interest to us. The formulation of the optimization problem is as follows.

Consider a linear functional $\Lambda_u$ defined on the space of functions of $\xi$, that is normalized to 1 on a particular conformal block, say the tilt operator $t_i$,

$$\Lambda_u((N-1)\xi^{\Delta_\phi} f_{\mathrm{bry}}(2, \xi)) = 1, \tag{5.1}$$

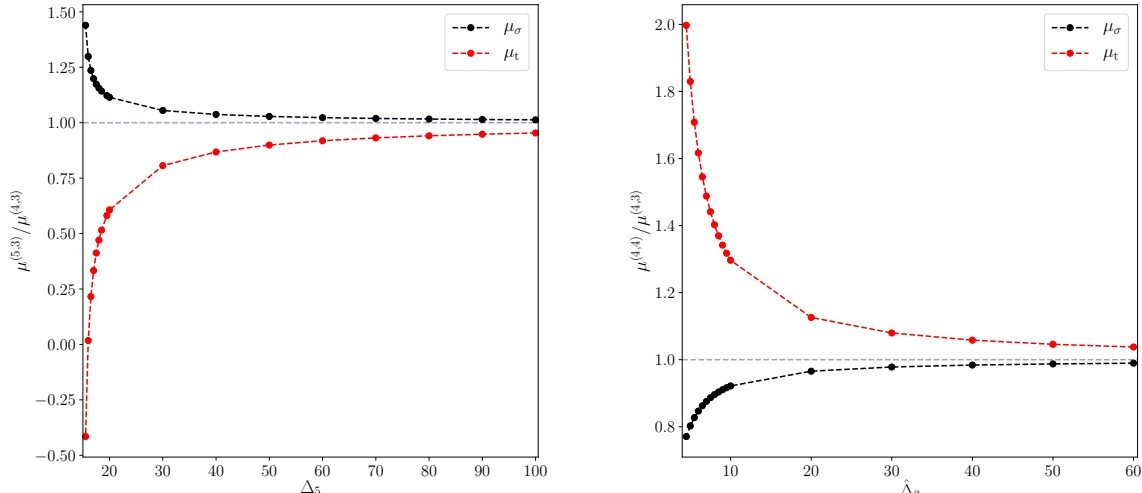

**Figure 6**: $\mu_\sigma$ and $\mu_t$ for $N = 2$ from the one parameter families $(5, 3)$ (left) and $(4, 4)$ (right) as functions of the dimension of the added operator in the respective channels. All values are normalized to the results from the $(4, 3)$ truncation so that the graphs tend to 1 as $\Delta_5, \hat{\Delta}_3 \to \infty$.

and simultaneously acts positively on all other blocks in the equation,

$$\Lambda_u(-N f_{\text{bulk}}(\Delta_k, \xi)) \geq 0,$$
$$\Lambda_u(\xi^{\Delta_\phi}) \geq 0,$$
$$\Lambda_u(\xi^{\Delta_\phi} f_{\text{bry}}(\widehat{\Delta}_n, \xi)) \geq 0, \tag{5.2}$$

for all allowed values of $\Delta_k \in [\Delta_{\min}, \infty)$ and $\hat{\Delta}_l \in [3, \infty)$. If such a functional exists, the crossing equation can only be satisfied if the OPE coefficient $\mu_t$ also obeys the upper bound

$$\mu_t \leq \Lambda_u(N). \tag{5.3}$$

The optimal bound in this case is found by minimizing $\Lambda_u(N)$. Notice that we can use the same logic to find lower bounds on OPE coefficients as well. Indeed, we already had the bounds $\mu_i \geq 0$, but we can place more stringent constraints on the OPE coefficients of the protected operators which are guaranteed to appear in the boundary OPE. To this end, we need to find another functional $\Lambda_l$ that is normalized to 1 on the same tilt operator block but acts negatively on all other blocks (considering $-N f_{\text{bulk}}(\Delta_k, \xi)$ to be the bulk blocks' contribution). Again, applying $\Lambda_l$ to the crossing equation gives the lower bound

$$\mu_t \geq \Lambda_l(N), \tag{5.4}$$

and the optimal lower bound is found by *maximizing* $\Lambda_l(N)$. Finding both the upper and the lower bound on the OPE coefficients $\mu_t$, $\mu_\sigma$ is crucial to determine the allowed range for their ratio, $\alpha(N)$.

As is customary in the numerical bootstrap literature, we choose the linear functionals $\Lambda_u$ and $\Lambda_l$ among the linear combinations of derivatives evaluated at $\xi = 1$:

$$\Lambda(f) = \sum_{m=0}^{M} a_m \, \partial_\xi^m f(\xi)\big|_{\xi=1} \, . \tag{5.5}$$

This is the same basis used earlier in our truncated bootstrap setup. In other words, we trade each conformal block for a $(M+1)$-dimensional vector:

$$-Nf_{\text{bulk}} \rightarrow \left(f_{\text{bulk}}^{(0)}, f_{\text{bulk}}^{(1)}, \ldots, f_{\text{bulk}}^{(M)}\right), \qquad \gamma\xi^{\Delta_\phi}f_{\text{bry}} \rightarrow \left(f_{\text{bry}}^{(0)}, f_{\text{bry}}^{(1)}, \ldots, f_{\text{bry}}^{(M)}\right), \qquad (5.6)$$

where $f_{\text{bulk}}^{(m)} \equiv -N\partial_\xi^m f_{\text{bulk}}\big|_{\xi=1}$ and $f_{\text{bry}}^{(m)} \equiv \gamma\partial_\xi^m(\xi^{\Delta_\phi}f_{\text{bry}})\big|_{\xi=1}$. The constant $\gamma$ is 1 for the displacement term and is $N-1$ for the tilt term in the crossing equation. For boundary operators in the unknown continuum $\hat{\Delta}_n > 3$, $\gamma$ was already absorbed in the definition of the OPE coefficients ($\tilde{\mu}_n \equiv \gamma\mu_n$) in eq. (2.22), but for the protected boundary operators it needs to be factored out explicitly.

Hereon, we use the usual semi-definite bootstrap setup to rewrite the optimization problem as a *polynomial program* which is the appropriate input to the SDPB package. A polynomial program is a rephrasing of the kind of optimization problems we have considered so far in which the vectors (5.6) are expressed as polynomials in $\Delta$, times a positive prefactor. As discovered in [70], powerful semi-definite programming methods can then be used to solve infinitely many constraints (for example, of the form of the first and last inequalities in (5.2) for a continuous interval of allowed $\Delta_k$) without discretization in $\Delta$.

The conformal blocks (2.6) and (2.7) are essentially hypergeometric functions.[10] The coefficients of their power-law expansions are rational functions of $\Delta$, which are easily turned into polynomials up to a positive prefactor. However, these expansions have unit radius of convergence, and therefore we cannot truncate them when we evaluate the blocks at $\xi = 1$. This problem was solved in [9] by expressing the bulk and boundary blocks in terms of new cross ratios, respectively $r$ and $\hat{r}$, defined as follows:[11]

$$r = \sqrt{\frac{2+\xi-2\sqrt{1+\xi}}{\xi}}, \qquad\qquad \hat{r} = 1 + 2\xi - 2\sqrt{\xi+\xi^2}. \qquad (5.7)$$

The $(r, \hat{r})$ coordinates have a nice geometric interpretation, which we do not dwell on here. Importantly, the point $\xi = 1$ corresponds to $r^2 = \hat{r} = 3-2\sqrt{2} < 1$, while the series expansions of the blocks converge up to $r^2$, $\hat{r} = 1$ and hence can be truncated. In fact, the blocks in the new coordinates turn out to still be hypergeometric functions in $r^2$ (see Appendix B). Finally, each term in (5.6) can be approximated as a polynomial in the corresponding scaling dimension by computing the series expansion up to a desired degree in $r^2$,

$$f_l^{(m)} \approx \chi_l(\Delta) \, P_l^{(m)}(\Delta). \qquad (5.8)$$

where $\chi_l$ is a strictly positive factor for all $\Delta$ and the subscript $l \in \{\text{bulk}, \text{bry}\}$ stands for one of the channels.

The problem of finding an upper bound for the OPE coefficient $\mu_{\text{t}}$, described above, is now restated as a polynomial program as follows. In the space of co-vectors $a_m \in \mathbb{R}^{M+1}$, we minimize $a_0$ such that

$$\sum_{m=0}^{M} a_m P_{\text{bry}}^{(m)}(2) = 1, \qquad (5.9)$$

---

[10]In fact, the boundary blocks $f_{\text{bry}}$ reduce to elementary functions in $d = 3$, see appendix B.

[11]We retain the Taylor series in $\xi$, so that the derivatives are still evaluated with respect to $\xi$ rather than $r$ and $\hat{r}$.

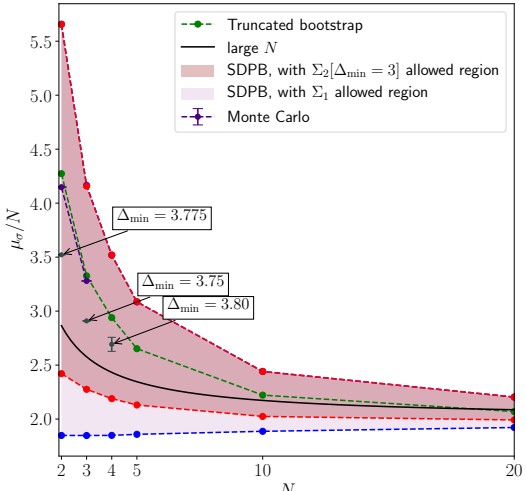 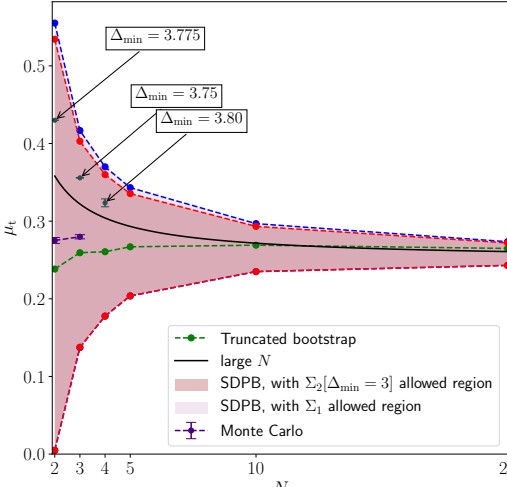

**Figure 7**: Bounds on the OPE coefficients $\mu_\sigma$ and $\mu_t$ from positive bootstrap, compared with truncated bootstrap, Monte Carlo (table 2) and perturbative results. The lightly shaded regions bounded between the blue lines represent the runs with $\Delta_{\min} = \Delta_\epsilon$, whereas the dark shaded regions between the red lines represent the stricter bounds from eq. (6.3). The solitary data points in grey are the improved lower bounds for $\mu_\sigma$ and upper bounds for $\mu_t$ for different $\Delta_{\min}$.

and

$$\sum_{m=0}^{M} a_m P_{\text{bulk}}^{(m)}(\Delta_k) \geq 0 \qquad\qquad \text{for } \Delta_k \in [\Delta_{\min}, \infty), \qquad\qquad (5.10)$$

$$\sum_{m=0}^{M} a_m P_{\text{bry}}^{(m)}(\hat{\Delta}_l) \geq 0 \qquad\qquad \text{for } \hat{\Delta}_l \in \{0\} \cup [3, \infty). \qquad\qquad (5.11)$$

In this incarnation, the problem is readily reinterpreted as an SDP and can be solved numerically to high precision. For a more detailed exposition on the mathematics behind `SDPB 2.0`, refer to [47, 69, 70]. Some technical details of the calculations specific to our implementation of the SDP for boundary bootstrap may be found in Appendix B.

## 6  Results

Having set up the boundary bootstrap problem in the language of positive bootstrap, we proceed to discuss the bounds obtained from the computation and compare them with previous estimates for $\alpha(N)$. Specifically, the positive bootstrap bounds in conjunction with the truncation method results provide very strong evidence of the existence of the extraordinary-log universality class for $N = 3$.

In our experience with the positive bootstrap for this problem, the key parameter is, unsurprisingly, the assumed gap in the bulk spectrum $\Delta_{\min}$. The more we assume about the theory, the less ways there are for sets of CFT data to satisfy crossing symmetry, which allows for tighter bounds on the

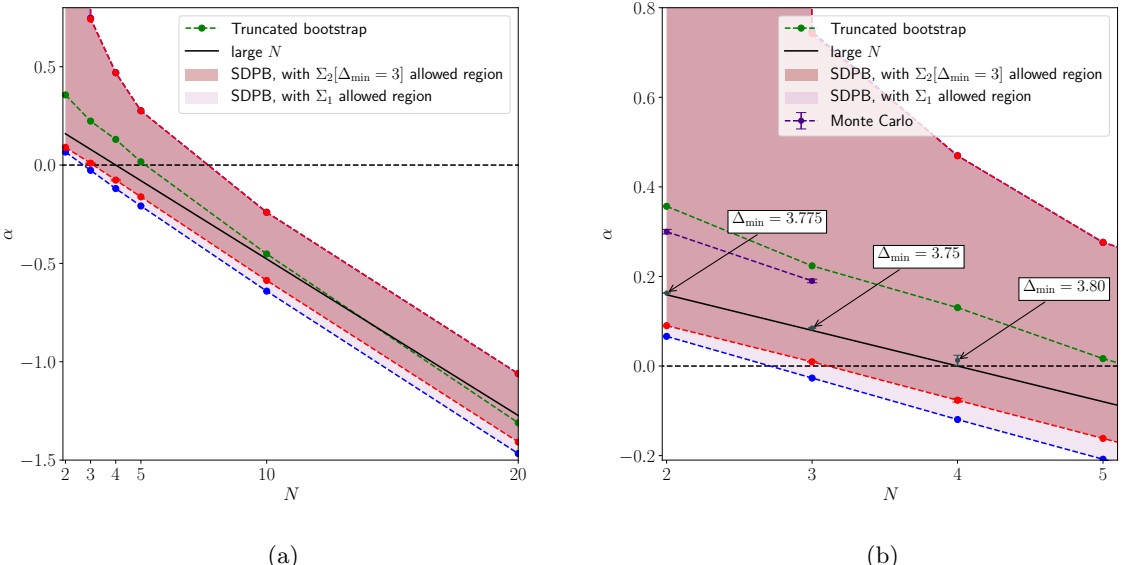

(a)                                    (b)

**Figure 8**: Various $\alpha(N)$ estimates compared. The convention for the shaded regions is the same as in figure 7. On the right, 8b is a zoomed version with more details around low values of $N$. It also highlights our extra runs with different $\Delta_{\min}$ values for $N = \{2, 3, 4\}$ as discussed in the text (shown in grey).

respective data. The most agnostic bounds were obtained by setting $\Delta_{\min} = \Delta_\epsilon$. We label this set of assumptions as $\Sigma_1$, for clarity in the discussion to follow:

$$\Sigma_1 \equiv \begin{cases} \Delta \in [\Delta_\epsilon, \infty), \\ \widehat{\Delta} \in \{0, 2\} \cup [3, \infty). \end{cases} \tag{6.1}$$

The bounds on the OPE coefficients $\mu_\sigma$, $\mu_t$, and $\alpha$ obtained from $\Sigma_1$ are represented by the lightly shaded regions in figures 7 and 8. With $\Sigma_1$, the bounds are so broad that they allow for $2 < N_c < 10$. The truncated bootstrap solution and the large $N$ solution are comfortably allowed within the bounds from $\Sigma_1$. In all our graphs/results, the bounds on $\alpha$ are calculated using the extremal values of $\mu_\sigma$ and $\mu_t$ allowed by bootstrap:

$$\frac{1}{32\pi} \frac{\mu_{\sigma,\min}}{\mu_{t,\max}} - \frac{N-2}{2\pi} \leq \alpha \leq \frac{1}{32\pi} \frac{\mu_{\sigma,\max}}{\mu_{t,\min}} - \frac{N-2}{2\pi}. \tag{6.2}$$

Now, we include the known operator $\epsilon$ as an *a priori* assumption in the bulk OPE along with its dimension. For $N \in \{2, 3, 4\}$, the scaling dimension of this operator, $\Delta_\epsilon$, is known to great precision with rigorous errors from previous bootstrap literature, reproduced earlier in table 1. We also assume that $\epsilon$ is the only relevant $O(N)$ singlet in the bulk OPE, or in other words for the next $O(N)$ singlet, $\Delta \geq \Delta_{\min} = 3$. To summarize, we have the new set of assumptions $\Sigma_2$ where

$$\Sigma_2[\Delta_{\min} = 3] \equiv \begin{cases} \Delta \in \{\Delta_\epsilon\} \cup [3, \infty), \\ \widehat{\Delta} \in \{0, 2\} \cup [3, \infty). \end{cases} \tag{6.3}$$

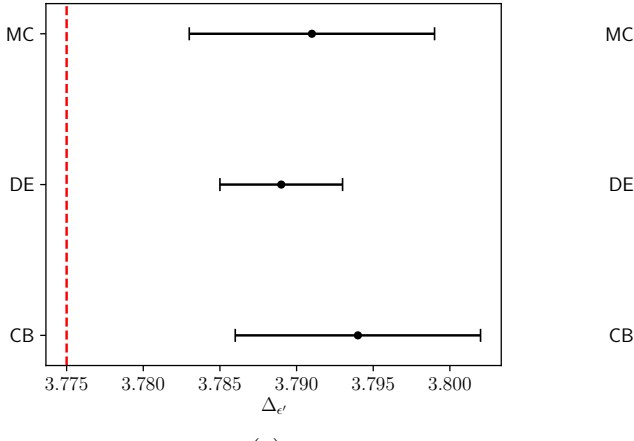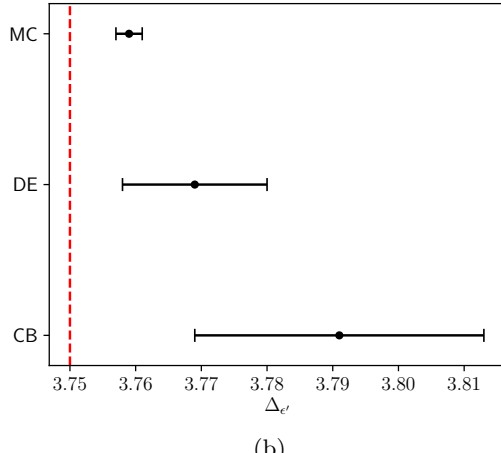

(a)                  (b)

**Figure 9**: Comparison of the $\Delta_{\epsilon'}$ estimates for the O(2) (left) and the O(3) (right) model from conformal bootstrap (CB), derivative expansion (DE) [58] and Monte Carlo (MC) methods. The CB estimates are from [52] and [71] for $N = 2$ and $N = 3$ respectively. The MC results are taken from refs. [72] and [54]. The dashed red lines mark the values we chose for the gap in the bulk spectrum above $\Delta_{\epsilon}$.

The new bounds show a concerted improvement in the lower bound for $\mu_\sigma$ and the upper bound for $\mu_{\rm t}$, and when combined they improve the lower bound on $\alpha$ just enough so that $\alpha(N = 3) \geq 0$ (dark shaded regions in figures 7 and 8). One can vary the input parameters $\Delta_\phi$ and $\Delta_\epsilon$ in the region allowed by the rigorous bootstrap bounds from literature to find the uncertainty in the lower bound for $\alpha$. The lower bound remains positive across the region, with

$$\alpha(3) \geq 0.00936(16). \tag{6.4}$$

We conclude that, under the assumptions that $\lambda_k > 0$ and that the boundary spectrum at the normal fixed point is gapped as in eq. (6.3),

*The* O(3) *universality class has an extraordinary-log boundary phase.*

The estimates for the scaling dimension of the next operator (which we have been calling $\epsilon'$), $\Delta_{\epsilon'}$, for the O(3) model in the literature vary depending on the methods used and no rigorous bootstrap results are available. However, it is possible to push $\Delta_{\min}$ safely to 3.75 (see figure 9b) in the interest of improving the lower bound on $\alpha(3)$. Doing so gives us

$$\alpha(3) \geq 0.08483(51), \tag{6.5}$$

which is closer to the value obtained from the truncated bootstrap, table 3, and Monte Carlo, table 2. Figure 8b shows this new estimate in relation to the previous ones. In addition, we also used $\Delta_{\min} = 3.775^{12}$ with the O(2) model to get the lower bound

$$\alpha(2) \geq 0.16294(15). \tag{6.6}$$

---

[12]See figure 9a for a visual comparison of estimates of $\Delta_{\epsilon'}$ in the literature.

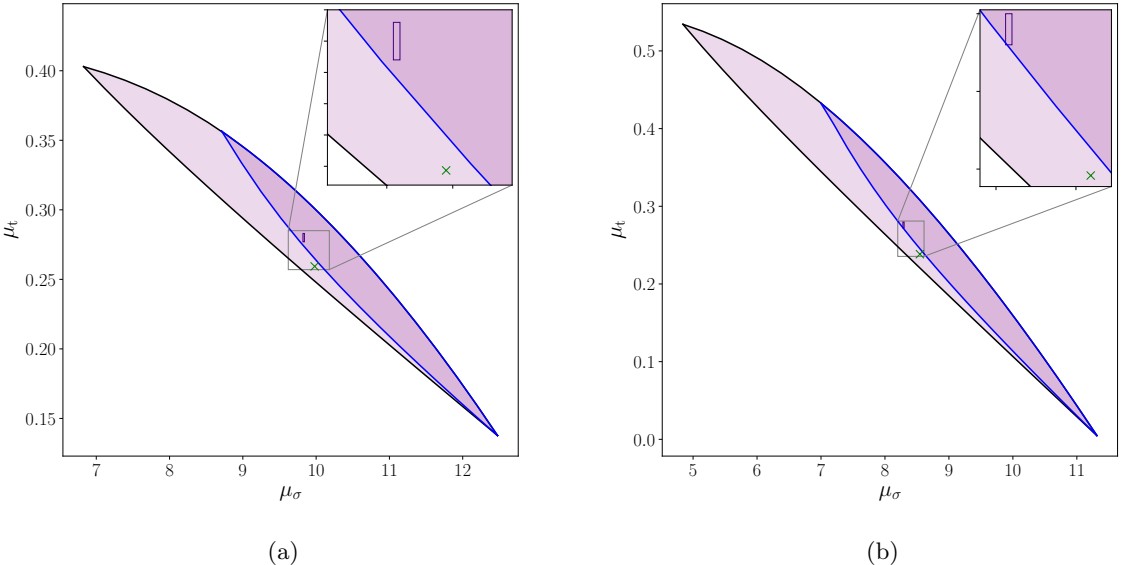

(a)                                                (b)

**Figure 10**: Allowed regions in the $(\mu_\sigma, \mu_t)$ coefficient space for $N = 3$ (left) and $N = 2$ (right). The bigger islands correspond to $\Delta_{\min} = 3$ and the smaller ones to a higher value of $\Delta_{\min}$, as discussed in the text. They are mapped by scanning over $\mu_\sigma$ and finding the bounds on $\mu_t$. Monte Carlo (purple rectangles) and truncated bootstrap (green crosses) results are included for comparison.

We note in passing that for $N = 2, 3$ Monte Carlo results for $\mu_\sigma$, $\mu_t$ in table 2 are safely within our tightest bounds obtained here.

One may wonder if the bounds on $\alpha$ presented in Fig. 8 can be improved without strengthening our assumptions. Consider for instance the lower bound: the values $\mu_{\sigma,\min}$ and $\mu_{t,\max}$ correspond in principle to *different* solutions to crossing, so they may not be reached at the same time. More generally, the positive solutions to crossing form a compact convex region in the $(\mu_\sigma, \mu_t)$ plane. All the results obtained so far only rely on the size of a rectangle which bounds this region. While this was sufficient to achieve the main goal of this work, it is interesting to carve out the shape of this island. This can be done by scanning over one coefficient while optimizing the other. At the technical level, we remove the positivity condition on $\xi^{\Delta_\phi}$ in (5.2) and impose, for instance, the upper bound

$$\mu_t \le \Lambda_u(N - \mu_{\sigma,\text{trial}} \, \xi^{\Delta_\phi}), \tag{6.7}$$

thus only optimizing among solutions to crossing where $\mu_\sigma = \mu_{\sigma,\text{trial}}$. The results for $N = 2$ and $N = 3$ are shown in figure 10. We notice sharp corners $(\mu_{\sigma,\min}, \mu_{t,\max})$ and $(\mu_{\sigma,\max}, \mu_{t,\min})$. This implies that, to a good approximation, the extremal values for $\alpha(N)$ are actually realized. Indeed, the optimal value for an OPE coefficient corresponds to a unique solution to crossing [66]. Let's consider, for instance, the solution corresponding to the maximum value of $\mu_t$, close to the upper corner of the island. If the corner is sharp, there is a unique value of $\mu_\sigma$ allowed at the tip, hence the solution to crossing contains both $\mu_{\sigma,\min}$ and $\mu_{t,\max}$. We conclude that we cannot improve our bounds further without changing our assumptions.[13] The same conclusion might be reached without mapping the

---

[13]For instance, one might impose the presence of the displacement and scan over its OPE coefficient, rather than

shape of the island, but rather looking at the extremal functionals [73], *i.e.* the functionals with maximal/minimal value on the identity block. Such functionals satisfy all the positivity conditions (5.2), and saturate some of them. They are useful because the scaling dimensions of the operators appearing in the solution to crossing are signaled by the zeros of the corresponding functional. We performed this check for the functionals that produce the $N = 2$ and $N = 3$ bounds. As expected, the two extremal functionals associated to $\mu_{\sigma,\min}$ and $\mu_{t,\max}$ respectively, have, with good precision, the same zeros. The same observation can be made for the functionals corresponding to $\mu_{\sigma,\max}$ and $\mu_{t,\min}$. This confirms again that the extremal values for $\alpha$ are realized, although the related solutions don't resemble the spectrum of the O($N$) model.

Figure 10 allows us to make a few other observations. As expected, increasing $\Delta_{\min}$ shrinks the islands leaving one corner invariant. The truncated bootstrap solutions lie barely outside of both of the smaller islands. Since the $(4, 3)$ truncated solution obeys the same assumptions on the spectrum which are used to generate the islands, the discrepancy is due to the systematic error of the Gliozzi method. Finally, notice that the Monte Carlo values lie quite close to the boundaries of the smaller islands. Let us mention that, varying the number of derivatives, the lower bounds of said islands shrink very slowly, and it is unlikely that the Monte Carlo values could turn out to be incompatible with positivity. Nevertheless, it would be interesting to push the numerics further in the future.

We now ask the following question: what is the minimal set of assumptions needed to prove that $\alpha(N = 4) > 0$ using bootstrap? Notice that the uncertainties in $\Delta_\phi$ and $\Delta_\epsilon$ are much larger for O(4) than for the O(2) and O(3) models. We approach this problem with two independent perspectives. For the first, we extend our analysis with the set $\Sigma_2[\Delta_{\min}]$ to allow for a variable $\Delta_{\min}$. Numerically, this approach boils down to finding the minimum $\Delta_{\min}$ in the assumption set $\Sigma_2[\Delta_{\min}]$ for which the lower bound on $\alpha$ is positive. Searching in steps of 0.1, the lowest value of $\Delta_{\min}$ that we found for which $\alpha(N = 4)$ is positive across the region of possible input parameters is $\Delta_{\min} = 3.80$.

In fact, the value $\Delta_{\min} = 3.80$ is curiously close to various estimates for the dimension of the first irrelevant O($N$) singlet, $\Delta_{\epsilon'}$.[14] We note that there is no reason why $\alpha$ should saturate the lower bound of our bootstrap results (for instance, saturation occurs neither when $N \to \infty$ nor for the $N = 2, 3$ Monte Carlo results). As we have previously discussed, the truncated bootstrap approach gives $N_c$ closer to 5.

In a complementary perspective, we scan over $\mu_\sigma$ in the interval allowed by our previous bounds with $\Sigma_2[\Delta_{\min} = 3]$ (6.3) to find better bounds on $\mu_t$. In other words, we do not require a large gap after $\Delta_\epsilon$ in the bulk spectrum. The lower bound on $\alpha$ turns out to be a monotonic function of $\mu_\sigma$ (figure 11), and we search for the *minimal* value of $\mu_\sigma$ for which $\alpha(4) > 0$. We obtain $\alpha(4) > 0$ if

$$\mu_\sigma \gtrsim 10.6. \tag{6.8}$$

Compare this to our estimate for the same from the truncated bootstrap, $\mu_{\sigma,\text{trunc}} = 11.758$, which safely satisfies the bound. The lowest lying OPE data (for instance, $\mu_\sigma$ in the boundary channel) are expected to be less sensitive to the truncation errors of Gliozzi's method. As discussed in section 4, this is confirmed by the Monte Carlo results for $N = 2, 3$ [41], which agree better with our $\mu_\sigma$ than with our $\mu_t$. It is reasonable to expect that $\mu_\sigma$ lies close to $\mu_{\sigma,\text{trunc}}$ for $N = 4$ as well. Our current analysis shows that it can be off by as much as 9% , and still $\alpha(4) > 0$, given positivity. Thus, we

---

simply allowing for its existence. And of course, the bounds might improve increasing $M$ in eq. (5.5)

[14]Table VII in Ref. [58] compiles this data. As far as we know, no conformal bootstrap results with rigorous errors exist. Ref. [71] estimates 3.817(30) using the bootstrap. Older Monte Carlo gives 3.765 [57] and very recent Monte Carlo gives 3.755(5) [49]. Derivative expansion gives 3.761(12) [58].

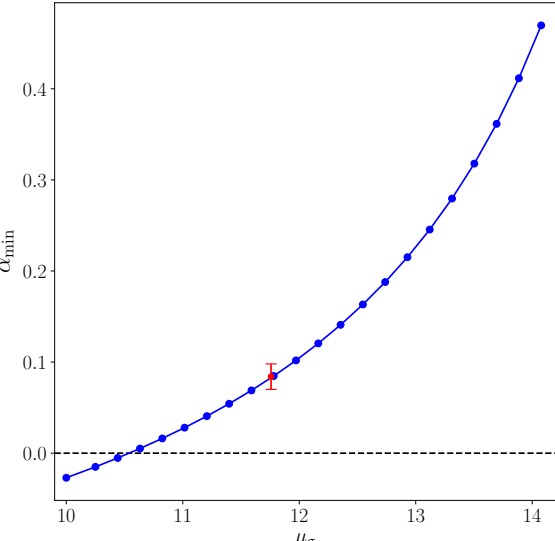

**Figure 11**: Lower bound on $\alpha(4)$ as a function of the imposed value of $\mu_\sigma$ (in eq. (6.7)), in an interval chosen so that it contains the zero of $\alpha_{\min}$. The red point and the errorbar correspond to $\mu_{\sigma,\text{trunc.}}$, which gives $\alpha(4) \geq 0.084(14)$. The reported error comes from the uncertainties in $\Delta_\phi$ and $\Delta_\epsilon$.

have found two independent minimal sets of assumptions which conclude that $N_c > 4$ and hence that the extraordinary-log phase is realized in the O(4) model.

We finally highlight a corollary of our analysis. The existence of a lower bound on $\mu_\sigma$ strictly larger than zero implies that the O($N$) symmetry is broken at the boundary. It is natural to ask what is the minimum set of assumptions on the boundary spectrum which implies this result. For all the values of $N$ in table 1, we checked that the mild assumption that there are no relevant operators in the boundary channel is sufficient.

Hence, we obtain the following rigorous result, which likely extends beyond the values of $N$ explicitly checked:

*All conformal boundary conditions for the* O($N$) *models with positive* $\lambda_k$ *and no relevant boundary operators break the global symmetry.*

Clearly, the positivity assumption lessens the scope of this result, but one may wonder if the stronger version of the same fact is true, namely that only symmetry breaking boundary conditions can be stable. We cannot provide evidence in either direction with the methods of this work.

## 7 Conclusions

The main purpose of this work was to identify the boundary critical behavior of O($N$) vector models in $d = 3$ by studying the normal fixed point using conformal bootstrap. In doing so, we rely on two important lines of inquiry, i.e. the boundary bootstrap program, and the scheme developed in [31] to study the stability of an extraordinary boundary phase starting from the normal fixed point. Our

target was the ratio of boundary OPE coefficients $\alpha(N)$ in eq. (3.7) and specifically the quantity $N_c$ at which $\alpha(N_c) = 0$.

We used two techniques at the forefront of the boundary bootstrap program, the truncated bootstrap in the spirit of Gliozzi, and the positive bootstrap using semi-definite programming. With the former method, we found exact zeros of the truncated crossing equation for the $(4, 3)$ truncation and estimated that $N_c \approx 5$. The truncated solution also provides estimates for the CFT data for various values of $N$, which are reported in table 3. It is encouraging that our results are reasonably close to the forthcoming Monte Carlo study [41] of the normal universality class in models with $N = 2$ and $N = 3$, summarized in table 2. Yet, the difficulty in quantifying the truncation error motivated us to look in the direction of the positive bootstrap.

Semi-definite programming allowed us to prove that crossing is only satisfied if $N_c > 3$, under the following three assumptions:

1. all the $\lambda_k$ are positive,

2. there is only one relevant O($N$) singlet in the bulk channel (namely, $\epsilon$),

3. at the normal fixed point, the only boundary operator with dimension less than 3 is the tilt, which has $\widehat{\Delta} = 2$.

The second assumption is of course true. We have compiled in appendix C results of the $2 + \epsilon$, large-$N$ and $4 - \epsilon$ expansions that are consistent with both assumption 1 and 3. Moreover, all the truncated solutions used to estimate the CFT data in this work are consistent with positivity of the bulk OPE coefficients.

Thus, the extraordinary-log phase and the special transition survive for Heisenberg magnets in $d = 3$. We also provide two scenarios under which one can claim $N_c > 4$ from positive bootstrap. The first scenario requires that the lightest irrelevant O($N$) singlet has $\Delta \geq 3.80$. An alternative sufficient condition is a stricter lower bound on the value of the OPE coefficient of the boundary identity, $\mu_\sigma \gtrsim 10.6$. Comparing to our previous estimate for this quantity from the truncated bootstrap, we deem high the likelihood that $N_c > 4$. The question of the evolution of the phase diagram for $N > N_c$ is not addressed in our paper and is the natural branching point from this work.

In the process of computing $\alpha(N)$, we obtained a number of other results. We presented the renormalization group argument of [31] under a slightly different perspective, which emphasizes the role of the Ward identities in shaping the action for the extraordinary phase. The main character in this story is the tilt operator: its coupling $C_t$ with the O($N$) current, as expressed by the OPE (2.15), determines $N_c$ through eq. (3.7). This is conceptually important, because $C_t$ appears in multiple OPEs and could in principle be measured independently of $\mu_\sigma$ and $\mu_t$. Some of the perturbative results reported in appendix C are also new. The $2 + \epsilon$ expansion has not appeared before, and we extracted new OPE data both at large $N$ and in $4 - \epsilon$ dimensions. In the latter case, we observed that the positivity conjecture can be turned into a surprisingly powerful algorithm to compute CFT data of the bulk CFT. For instance, the leading contribution to the three-point function $\langle \phi\phi\phi^{2n} \rangle$, which is of order $\epsilon^{n-1}$, can be computed from the knowledge of a correlator at $O(\epsilon)$ at the normal fixed point. On the numerical bootstrap side, we showed that, under the assumptions listed above, the O($N$) order parameter in the bulk necessarily has a non-zero expectation value, if no relevant ($\widehat{\Delta} < 2$) operators appear in the boundary OPE.

Furthermore, we explored the stability of the solutions obtained from the truncated bootstrap. While we have not settled the issue of bounding the systematic errors from truncation, this work

showcases the gap in the literature on this question. Solving it will have implications beyond boundary bootstrap and will open up the conformal bootstrap to important problems in statistical mechanics that are well-"known" non-unitary CFTs, such as Anderson transitions and turbulence. In this context, it is worth pointing out that reference [66] offers a way to use semi-definite programming for a class of non-positive solutions to crossing, so-called extremal solutions. It would be interesting to explore applications of this method, for instance to boundary bootstrap.

Our work also provides an impetus for improving the data on higher singlet operators $\{\Delta_\epsilon, \Delta_{\epsilon'}\}$ which appear in the $\phi \times \phi$ OPE for the O($N$) vector models, especially for $N \geq 4$. In the future, it would also be interesting to include in the analysis the bulk operators transforming in traceless symmetric representations of O($N$). This requires using both the crossing constraints on $G_\sigma$ and $G_\phi$ in eqs. (2.20) and (2.21). In particular, it would be nice to check perturbatively if all $\lambda_l$'s in each equation have the same sign, which would motivate the use of SDPB. In the $4 - \epsilon$ expansion, this can be done using the results in [13]. It is worth remarking that the upper bound on $\alpha$ improved very little, if at all, changing the assumptions on the bulk singlet spectrum, see figures 7 and 8. One may wonder if the inclusion of the traceless symmetric sector can make a difference.

In this paper, we have focused on O($N$) models with integer $N \geq 2$. It will be interesting to extend the discussion to non-integer $N$ (including the range $N < 2$), where the O($N$) model can be defined on the lattice as a loop model. CFT with O($N$) symmetry has been extensively studied in $d = 2$ (see, for example, a recent paper [74] and references therein), including the surface critical behavior and the extraordinary transition [75]. It seems natural and worthwhile to revisit boundary criticality in models with $N < 2$ in $d = 3$ and other dimensions.

Let us comment on the prospects of observing the $N = 3$ extraordinary-log universality class in experiments. This requires two conditions to be met. First, the material should have sufficiently strong magnetic exchange on the boundary compared to the bulk. (In the classical O(3) model (1.1) the critical $K_1/K \approx 1.8$ [36].) Such an enhancement might occur naturally or one may attempt to engineer it by depositing a material with a higher $T_c$ on the surface. Second, the spin-orbit coupling should be very weak, as any anisotropy of the O(3) order parameter is a very relevant perturbation in the extraordinary-log phase. (This is in contrast to the cubic anisotropy in the bulk, which is very weakly relevant [53].) Under these ideal conditions, we expect the surface magnetization to onset very sharply below the bulk $T_c$ as

$$m_s \sim [\log(T_c - T)]^{-q/2} . \tag{7.1}$$

This should be compared to the more gradual onset of the bulk magnetization $m_b \sim (T_c - T)^\beta$, $\beta \approx 0.37$ [53], or of the surface magnetization for the ordinary boundary universality class $m_s \sim (T_c - T)^{\beta_s^o}$, $\beta_s^o \approx 0.85$ [36]. In practice, the logarithm in (7.1) might be difficult to observe and the surface magnetization will appear to jump to a finite value below $T_c$. A small magnetic exchange anisotropy is expected to split the surface transition temperature from the bulk $T_c$, giving rise to a thin sliver of surface-ordered (or quasi-long-range ordered in the case of XY anisotropy)/bulk-disordered phase as in figure 1.

## Acknowledgments

We are very grateful to Francesco Parisen Toldin for sharing his results prior to publication and for insightful comments on the manuscript. Marco Meineri would like to thank Madalena Lemos and Tobias Hansen for useful discussions. The authors are grateful to John Cardy, Ferdinando Gliozzi, Tobias Hansen, Zohar Komargodski, Madalena Lemos, Miguel Paulos, Joao Penedones, Balt van Rees

and Slava Rychkov for their comments on the draft, and to Johan Henriksson for finding a typo in appendix C and for pointing out refs. [76, 77]. The authors would also like to thank the organizers and the participants of the Bootstat conference held at the Institut Pascal in May 2021, for useful discussions on topics related to this project. Max Metlitski is supported by the National Science Foundation under grant number DMR-1847861. Marco Meineri is supported by the Swiss National Science Foundation through the Ambizione grant number 193472. The work of AK was supported by the National Science Foundation Graduate Research Fellowship under Grant No. 1745302. AK also acknowledges support from the Paul and Daisy Soros Fellowship and the Barry M. Goldwater Scholarship Foundation. The authors acknowledge the use of the Unity high-performance computing cluster at the College of Arts and Sciences, the Ohio State University and the Owens cluster at Ohio Supercomputing Center [78] made available for conducting the research reported in this paper.

## A    Ward identities and the tilt operator

This appendix is dedicated to a review of some consequences of the Ward identity (2.16) that defines the tilt operator. We mostly keep the dimension of space generic, and we specify $d = 3$ when making contact with the setup of subsection 2.2. In general, every continuous symmetry[15] broken by a conformal defect corresponds a protected boundary operator t. If we denote by $\mathcal{D}$ the submanifold where the defect is located (e.g. $x^3 = 0$ in this work) and by $\delta_{\mathcal{D}}$ the delta function with support on the said submanifold, then the tilt operator is defined by the following contact term:

$$\partial_\mu j^\mu(x) = \delta_{\mathcal{D}}(x)\sqrt{C_{\mathrm{t}}}\,\mathrm{t}(x), \tag{A.1}$$

where $j^\mu$ is the current associated to the broken symmetry. This equation fixes the scaling dimension of the tilt, $\widehat{\Delta}_{\mathrm{t}}$, to be the dimension of the defect. As usual, a topological operator can be constructed from the flux of the current:

$$Q(\Sigma) = \int_\Sigma \star j. \tag{A.2}$$

The integral is over a co-dimension one surface $\Sigma$. Now consider the correlation function of $Q(\Sigma)$ with a bulk primary $\mathcal{O}$, and choose $\Sigma$ such that it separates the local operator from the defect, as in figure 12. We can compute this correlator in two ways: either we deform $\Sigma$ towards the left, picking up the contact term in eq. (A.1), or towards the right, where we can use the usual Ward identity:

$$\partial_\mu j^\mu(x)\,\mathcal{O}(y) = \delta^d(x - y)\,\delta\mathcal{O}(y), \tag{A.3}$$

where $\delta\mathcal{O}$ denotes the variation of $\mathcal{O}$ under the action of the generator of the symmetry. We then readily obtain the equality

$$\sqrt{C_{\mathrm{t}}}\,\langle\left(\int_{\mathcal{D}}\mathrm{t}\right)\mathcal{O}(y)\rangle + \langle\delta\mathcal{O}(y)\rangle = 0. \tag{A.4}$$

Since both one-point functions and correlators of one bulk and one defect operator are fixed by conformal symmetry up to OPE coefficients, eq. (A.4) provides one relation for the OPE data. We are interested in the special case of a codimension one defect, with $\mathcal{O} = \phi$ being a scalar, in which case the correlation functions involved read

$$\langle\mathrm{t}(0)\phi(y)\rangle = \frac{b_{\mathrm{t}}}{(2y^d)^{\Delta_\phi - d + 1}(y^2)^{(d-1)}}, \qquad \langle\delta\phi(y)\rangle = \frac{\delta a}{(2y^d)^{\Delta_\phi}}. \tag{A.5}$$

---

[15]The extension to higher spin symmetries is straightforward

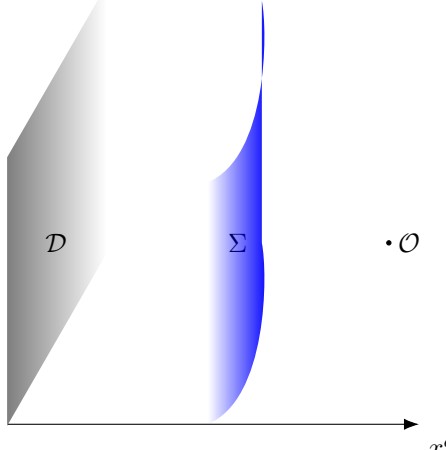

**Figure 12**: Configuration of the defect $\mathcal{D}$, the topological operator $Q(\Sigma)$ and the local operator $\mathcal{O}$ which yields the Ward identities (A.6).

Here we used the fact that the dimension of t is fixed to $\widehat{\Delta}_\text{t} = d - 1$ by eq. (A.1). Furthermore, $a$ (or in this case, $\delta a$) and $b$ are the same OPE coefficients appearing in (2.4). Plugging this into eq. (A.4), we get

$$\frac{\delta a}{b_\text{t}} = -\frac{2\pi^{d/2}}{\Gamma(d/2)}\sqrt{C_\text{t}}. \tag{A.6}$$

Let us now specify this formula to the case of interest, where eq. (A.1) becomes

$$\partial_\mu j^\mu_{[Ni]} = \delta(x^d)\sqrt{C_\text{t}}\,\text{t}_i. \tag{A.7}$$

The multiplet of tilt operators lives in the coset $\text{O}(N)/\text{O}(N-1)$, and therefore transforms as a vector under the preserved $\text{O}(N-1)$ subgroup. Hence, the only non trivial information is obtained by taking $\phi \to \phi_j$ in eq. (A.5). The variation then is[16]

$$\delta_{[Ni]}\phi_j = \delta_{Nj}\phi_i - \delta_{ij}\phi_N = -\delta_{ij}\,\sigma. \tag{A.8}$$

Going back to eq. (A.6), we find

$$\frac{a_\sigma}{b_\text{t}} = \frac{2\pi^{d/2}}{\Gamma(d/2)}\sqrt{C_\text{t}}, \tag{A.9}$$

which, specified to $d = 3$ and squared, is eq. (2.14).

## B  Technical details and numerical bounds from SDPB

This appendix includes miscellaneous details and results from our implementation of semi-definite programming for the positive bootstrap.

---

[16]Unfortunately, the symbols for a variation and for the Kronecher delta are conventionally the same: we hope this equation is clear nevertheless.

## B.1 Conformal blocks

For the positive bootstrap, we use the conformal blocks in $r$ and $\hat{r}$ coordinates, referenced in (5.7) and reproduced here for convenience:

$$r = \sqrt{\frac{2 + \xi - 2\sqrt{1 + \xi}}{\xi}}, \qquad\qquad \hat{r} = 1 + 2\xi - 2\sqrt{\xi + \xi^2}. \tag{B.1}$$

To get the bulk conformal block, one solves the Casimir equation for the $SO(d+1,1)$ conformal group in $r$ coordinates, which gives (in $d = 3$),

$$f_{\text{bulk}}(r, \Delta) = \frac{(2r)^\Delta}{1 + r^2} \, _2F_1\left(\frac{1}{2}, \Delta - 1, \Delta - \frac{1}{2}, r^2\right). \tag{B.2}$$

Manifestly, the conformal block is a hypergeometric function times a positive prefactor, and the hypergeometric function admits a power series expansion which is truncable at $r^2(\xi = 1) \approx 0.17$. For the boundary blocks, we use the other cross-ratio, $\hat{r}$ which produces a simple positive definite function in 3 dimensions:

$$f_{\text{bry}}(\hat{r}, \hat{\Delta}) = \frac{(4\hat{r})^{\hat{\Delta}}}{1 - \hat{r}^2}. \tag{B.3}$$

Let us describe briefly the procedure used to compute the polynomial approximations for the bulk blocks. Firstly, one can write the derivatives $d^n/d\xi^n$ in terms of $d^n/dr^n$ by means of the chain rule and its higher order generalizations. The problem now reduces to finding approximations of (B.2) and its derivatives which are polynomials in $\Delta$. The series expansion of $f_{\text{bulk}}$ is taken upto a desired degree $O(r^{n_r})$. The coefficients of this series are ratios of polynomials in $\Delta$, but the denominators are safely extracted out as a prefactor. For a given order $n_r$, the prefactor is

$$\chi_{\text{bulk}}(\Delta) = (2r)^\Delta \prod_{\Delta_* = -1/2}^{n_r/2 - 1} \frac{1}{(\Delta - \Delta_*)}, \tag{B.4}$$

where $\Delta_*$ goes over half-integer values in the given range. Factoring out these same poles, and substituting $r \to r(\xi = 1)$, we obtain the polynomial approximation for each derivative of the bulk block, upto $M$ derivatives.

## B.2 Parameters

A few more parameters are needed to fully describe the semi-definite programming calculations we did with SDPB 2.0. The bulk conformal blocks were approximated upto $n_r = 50$ degree in $r$. We also chose $M = 17$ number of derivatives to define the space of linear functionals to be optimized in. All calculations were done with a precision of prec $= 700$. The polynomial approximations were calculated in Mathematica and exported in XML files as input to SDPB 2.0.

## B.3 Numerical values of the bounds

For reference, we also provide the numerical values of the bounds in figure 7 obtained from the positive bootstrap. Table 4 lists the bounds for the boundary OPE coefficients $\mu_\sigma$ and $\mu_t$ with the assumptions $\Sigma_2[\Delta_{\min} = 3]$. Let us pause here to dwell on the computation of the errors on the bounds from positive bootstrap presented in this work. Both here and in the main text, the quoted errors are from the input parameters. We compute the bound on a $7 \times 7$ grid in $(\Delta_\phi, \Delta_\epsilon)$ that covers the region of allowed

values as per the literature. We found that the regions of interest are small, and the CFT data are essentially featureless inside. The coarse grid is enough to find the range of variation in the bound values. We only calculate the error on the bounds in cases that are relevant to the main message of this work.

| $N$ | $(\mu_{\sigma,\min}, \mu_{\sigma,\max})$ | $(\mu_{t,\min}, \mu_{t,\max})$ |
|---|---|---|
| 2 | (4.841, 11.313) | $(4.897 \times 10^{-3}, 0.534)$ |
| 3 | (6.8273(39), 12.475) | (0.138, 0.40298(15)) |
| 4 | (8.759(80), 14.077) | (0.178, 0.3600(29)) |
| 5 | (10.653, 15.432) | (0.204, 0.335) |
| 10 | (20.246, 24.41) | (0.235, 0.293) |
| 20 | (39.837, 44.078) | (0.243, 0.272) |

**Table 4**: Bounds on the boundary OPE coefficients from the positive bootstrap.

Finally, we did some solitary calculations to improve the lower bounds on $\alpha$, as discussed in Sec 6. The bounds thus produced are reported separately in table 5.

| $N$ | $\Delta_{\min}$ | $\mu_{\sigma,\min}$ | $\mu_{t,\max}$ |
|---|---|---|---|
| 2 | 3.775 | 7.0426(45) | 0.42995(18) |
| 3 | 3.75 | 8.727(13) | 0.35579(31) |
| 4 | 3.80 | 10.77(26) | 0.3236(51) |

**Table 5**: Additional bounds from the positive bootstrap

## C  Perturbative results

This appendix is dedicated to various perturbative results for the normal fixed point.

### C.1  2+$\epsilon$ expansion

In this subsection we use the $2 + \epsilon$ expansion to compute $\langle \phi^a(x)\phi^a(y)\rangle$[17] at the normal fixed point to order $\epsilon^2$. Boundary criticality in nonlinear sigma models has been studied in $2 + \epsilon$ dimensions at the ordinary transition [79], but to our knowledge the present work is the first utilization of such a model to study the normal fixed point. Although this is a conventional computation, we present it in some detail. The correlator to order $\epsilon$ has previously appeared in [80].

We begin with the non-linear $\sigma$-model for the field $\vec{\phi}$ in $d = 2 + \epsilon$ with a codimension one boundary:[18]

$$L = \frac{1}{2g}(\partial_\mu \vec{\phi})^2, \quad \vec{\phi}^2 = 1. \tag{C.1}$$

---

[17]Summation over repeated indices is understood in this section.

[18]This should be distinguished from eq. (3.5) in section 3 where the non-linear sigma model is two dimensional and lives on the boundary of $3d$ space.

We write $\vec{\phi} = (\vec{\pi}, \sqrt{1 - \vec{\pi}^2})$ so that

$$L = \frac{1}{2g} \left[ (\partial_\mu \vec{\pi})^2 + \frac{1}{1 - \vec{\pi}^2} (\vec{\pi} \cdot \partial_\mu \vec{\pi})^2 \right]. \tag{C.2}$$

We use dimensional regularization. We have $g = \mu^{-\epsilon} g_r Z_g(g_r)$, $\vec{\phi} = Z_\phi^{1/2} \vec{\phi}_r$ with [81]

$$Z_g(g_r) \approx 1 + \frac{(N-2)\tilde{g}_r}{\epsilon}, \quad Z_\phi \approx 1 + \frac{(N-1)\tilde{g}_r}{\epsilon}, \tag{C.3}$$

$$\beta(\tilde{g}_r) \approx \epsilon \tilde{g}_r - (N-2)\tilde{g}_r^2(1 + \tilde{g}_r), \qquad \Delta_\phi \approx \frac{\epsilon}{2} \frac{N-1}{N-2} \left( 1 - \frac{\epsilon}{N-2} \right), \tag{C.4}$$

and

$$\tilde{g}_r = g_r N_d, \qquad N_d = \frac{2}{(4\pi)^{d/2}\Gamma(d/2)}. \tag{C.5}$$

We first fix the bulk normalization of the field $\vec{\phi}$. We let $\vec{\phi}_{nrm} = C\vec{\phi}_r$ and demand that $\langle \phi_{nrm}^a(x)\phi_{nrm}^a(y) \rangle = \frac{N}{(x-y)^{2\Delta_\phi}}$. In the absence of a boundary, we have the propagator

$$\langle \pi^i(x)\pi^j(0) \rangle_0 = \delta^{ij} g D_0(x), \qquad D_0(x) = \frac{c_d}{x^{d-2}}, \qquad c_d = \frac{\Gamma(d/2-1)}{4\pi^{d/2}}. \tag{C.6}$$

Then, the bare field correlation function is

$$\langle \phi^a(x)\phi^a(0) \rangle \approx 1 - \langle \vec{\pi}^2 \rangle + \langle \pi^i(x)\pi^i(0) \rangle = 1 + (N-1)g D_0(x) \approx 1 + \frac{(N-1)g_r}{2\pi\epsilon} \left( 1 - \frac{\epsilon}{2}\gamma_E - \frac{\epsilon}{2}\log\pi - \epsilon\log\mu x \right). \tag{C.7}$$

So

$$\langle \phi_r^a(x)\phi_r^a(0) \rangle \approx 1 + (N-1)\tilde{g}_r(\log 2 - \gamma_E - \log\mu x). \tag{C.8}$$

Thus, after inserting the fixed point value $\tilde{g}_r^* \approx \frac{\epsilon}{N-2}$, we obtain

$$C \approx \sqrt{N} \mu^{\Delta_\phi} (1 - \Delta_\phi(\log 2 - \gamma_E)). \tag{C.9}$$

We next proceed to the system in the presence of a boundary. To avoid clutter, expectation values are denoted with the same symbol $\langle \ldots \rangle$, but from now on the presence of the boundary is understood. We access the normal universality class by imposing Dirichlet boundary conditions $\vec{\pi}(x^d = 0) = 0$. Now the free $\pi$ propagator is

$$\langle \pi^i(x)\pi^j(y) \rangle_0 = \delta^{ij} g D_d(x,y), \qquad D_d(x,y) = D_0(x-y) - D_0(x-Ry), \tag{C.10}$$

where $R(\vec{x}, x^d) = (\vec{x}, -x^d)$. We now compute the transverse and longitudinal two point functions to leading non-trivial order in $\epsilon$. We note that the connected longitudinal correlation function $\langle \phi^N(x)\phi^N(y) \rangle_{conn}$ only starts at $O(\epsilon^2)$ so we compute the disconnected longitudinal components first:

$$\langle \phi^N(x) \rangle = 1 - \frac{1}{2}\langle \vec{\pi}^2 \rangle = 1 - \frac{g(N-1)}{2} D_d(x,x) = 1 + \frac{g_r(N-1)}{4\pi\epsilon} \left( 1 - \frac{\epsilon\gamma_E}{2} - \frac{\epsilon}{2}\log\pi - \epsilon\log(2x_d) \right). \tag{C.11}$$

After multiplying by $Z_\phi^{-1/2}$ and $C$, and setting $g_r$ to its fixed point value, we obtain:

$$\langle \phi_{nrm}^N(x) \rangle = \frac{a_\sigma}{(2x^d)^{\Delta_\phi}}, \quad \mu_\sigma = a_\sigma^2 = N + O(\epsilon^2). \tag{C.12}$$

To $O(\epsilon)$, the longitudinal two point function is $\langle \phi^N_{nrm}(x) \rangle \langle \phi^N_{nrm}(y) \rangle$. We compute the connected correlation function and $\mu_\sigma$ to $O(\epsilon^2)$ later in this section. We also determine $\mu_\sigma$ to $O(\epsilon^2)$ below.

From Eq. (C.10), the transverse correlation function to $O(\epsilon)$ is

$$\langle \pi^i_{nrm}(x) \pi^j_{nrm}(y) \rangle = \delta^{ij} \frac{N\epsilon}{2(N-2)} \mu^{2\Delta_\phi} \log\left(\frac{1+\xi}{\xi}\right) \approx \delta^{ij} \mu_t \frac{1}{(x-y)^{2\Delta_\phi}} \xi^{\Delta_\phi} f_{\text{bry}}(1,\xi), \qquad \text{(C.13)}$$

with $\mu_t \approx \frac{N\epsilon}{2(N-2)}$. (We will be able to determine $\mu_t$ to $O(\epsilon^2)$ below.) Thus, the transverse correlation function is saturated to leading order by the boundary conformal block of the tilt operator with $\Delta_t = d - 1 \approx 1$. Combining the transverse and longitudinal contributions,

$$\langle \phi^a_{nrm}(x) \phi^a_{nrm}(y) \rangle = \frac{N}{(x-y)^{2\Delta_\phi}}(1 + \Delta_\phi \log(1+\xi)). \qquad \text{(C.14)}$$

Decomposing this into bulk conformal blocks, we find that the correlator is saturated by just one operator with dimension $\Delta \approx 2$ and

$$\lambda_{\Delta=2} \approx \Delta_\phi + O(\epsilon^2). \qquad \text{(C.15)}$$

This operator is the single relevant $O(N)$ singlet of the $O(N)$ model. Note that $\lambda_{\Delta=2}$ is positive in accord with the conjecture in section 5.

We now proceed to next order in $\epsilon$. We denote the first correction to the transverse correlator by $\langle \pi^i(x)\pi^j(y) \rangle_1 = \delta^{ij} D^1_\pi(x,y)$. Then

$$D^1_\pi(x,y) = -g^2 \int d^d w \left[ D_d(x,w) D_d(w,y) \lim_{w' \to w} \partial^w_\mu \partial^{w'}_\mu D_d(w,w') + \partial^w_\mu D_d(x,w) \partial^w_\mu D_d(w,y) D_d(w,w) \right.$$

$$\left. + N \left( \partial^w_\mu D_d(x,w) D_d(w,y) + D_d(x,w) \partial^w_\mu D_d(w,y) \right) \lim_{w' \to w} \partial^w_\mu D_d(w,w') \right]. \qquad \text{(C.16)}$$

Integrating by parts, we obtain:

$$D^1_\pi(x,y) = -g^2 \left( \int d^d w \left[ D_d(x,w) D_d(w,y) \left( \lim_{w' \to w} \partial^w_\mu \partial^{w'}_\mu D_d(w,w') - N \partial^w_\mu \lim_{w' \to w} \partial^w_\mu D_d(w,w') \right) \right. \right.$$

$$\left. - D_d(x,w) \partial^w_\mu D_d(w,y) \partial^w_\mu \left( \lim_{w' \to w} D(w,w') \right) \right] + D_d(x,y) D_d(y,y) \Bigg)$$

$$= -g^2 c_d \left( 2(d-2) \int d^d w \left[ D_d(x,w) D_d(w,y) \frac{(N-1)(d-1)}{(2w^d)^d} - D_d(x,w) \frac{\partial}{\partial w^d} D_d(w,y) \frac{1}{(2w^d)^{d-1}} \right] \right.$$

$$\left. - \frac{1}{(2y^d)^{d-2}} D_d(x,y) \right), \qquad \text{(C.17)}$$

where $c_d = \frac{\Gamma(d/2-1)}{4\pi^{d/2}}$. While the integral above can be taken explicitly (in particular, by going to momentum space in the direction along the boundary), here we use a different approach. Let's apply $-\partial^2_x$ to $D^1_\pi(x,y)$:

$$-\partial^2_x D^1_\pi(x,y) = g^2 c_d \left( \frac{1}{(2x^d)^{d-2}} \delta^d(x-y) - 2\frac{(N-1)(d-2)(d-1)}{(2x^d)^d} D_d(x,y) + \frac{2(d-2)}{(2x^d)^{d-1}} \frac{\partial}{\partial x^d} D_d(x,y) \right)$$

$$\approx \frac{g^2 \mu^\epsilon}{2\pi} \left[ \left( \frac{1}{\epsilon} - \log(2\mu x^d) - \frac{\gamma_E}{2} - \frac{\log \pi}{2} \right) \delta^d(x-y) - \frac{N-1}{2(x^d)^2} D_d(x,y) + \frac{1}{x^d} \frac{\partial}{\partial x^d} D_d(x,y) \right]. \qquad \text{(C.18)}$$

We write

$$D^1_\pi(x,y) = \frac{g^2\mu^\epsilon}{2\pi\epsilon}\left(1 - \frac{\epsilon}{2}(\log(4\mu^2 x^d y^d) + \gamma_E + \log\pi)\right)D_d(x,y) + D_c(x,y), \tag{C.19}$$

where $D_c(x,y)$ satisfies

$$\partial_x^2 D_c(x,y) = \frac{g^2\mu^\epsilon(N-2)}{4\pi(x^d)^2}D_d(x,y) \approx \frac{g_r^2(N-2)}{(4\pi x^d)^2}\log\left(\frac{\xi+1}{\xi}\right). \tag{C.20}$$

We use an ansatz $D_c(x,y) = D_c(\xi)$, then for $d \to 2$, $\partial_x^2 = \frac{1}{(x^d)^2}\left(\xi(\xi+1)\frac{\partial^2}{\partial\xi^2} + (2\xi+1)\frac{\partial}{\partial\xi}\right)$, so

$$\xi(\xi+1)D_c''(\xi) + (2\xi+1)D_c'(\xi) = \frac{g_r^2(N-2)}{(4\pi)^2}\log\frac{\xi+1}{\xi}. \tag{C.21}$$

Integrating the above equation,

$$D_c(\xi) = -\frac{g_r^2(N-2)}{(4\pi)^2}\left(\log\xi\log(\xi+1) + 2\text{Li}_2(-\xi) + \frac{\pi^2}{3}\right), \tag{C.22}$$

where the two integration constants are fixed so that $D_c(\xi) \to 0$ as $\xi \to \infty$ and so that $D_c(\xi)$ has no logarithmic divergence as $\xi \to 0$ (such a logarithmic divergence would lead to a $\delta^d(x-y)$-term in $\partial_x^2 D_c(x,y)$, which is not present on the right-hand-side of Eq. (C.20)).

Combining $D^1_\pi(x,y)$ with the zeroth order contribution to $\langle\pi^i(x)\pi^j(y)\rangle$, and expressing $g$ in terms of $g_r$, multiplying by $Z_\phi^{-1}$ and $C^2$, and using the fixed point value $\tilde{g}_r^* \approx \frac{\epsilon}{N-2}\left(1 - \frac{\epsilon}{N-2}\right)$, we obtain:

$$\langle\pi^i_{nrm}(x)\pi^j_{nrm}(y)\rangle$$
$$\approx \frac{\delta^{ij}\mu_t}{(x-y)^{2\Delta_\phi}}\xi^{\Delta_\phi}\left[\log\left(\frac{\xi+1}{\xi}\right) + \epsilon\left(\log\left(\frac{\xi+1}{\xi}\right)(1-\log\xi) - \frac{1}{4}\log^2\left(\frac{\xi+1}{\xi}\right) + \text{Li}_2(-1/\xi)\right)\right]$$
$$\approx \frac{\delta^{ij}\mu_t}{(x-y)^{2\Delta_\phi}}\xi^{\Delta_\phi}f_{\text{bry}}(1+\epsilon,\xi) ,$$

with

$$\mu_t \approx \frac{\epsilon N}{2(N-2)}\left(1 - \epsilon\frac{N-1}{N-2}\right). \tag{C.23}$$

Thus, to this order in $\epsilon$ the transverse correlator is still fully saturated by the boundary conformal block of the tilt operator with dimension $\hat{\Delta}_t = 1 + \epsilon$, as can be checked by expanding the conformal block in $\epsilon$.

We next proceed to the correction to the longitudinal correlator. We have the connected two-point function

$$\langle\phi^N(x)\phi^N(y)\rangle_{conn} \approx \frac{1}{4}\langle\vec{\pi}^2(x)\vec{\pi}^2(y)\rangle_{conn} = \frac{N-1}{2}g^2 D_d(x,y)^2 \approx \frac{(N-1)g_r^2}{32\pi^2}\log^2\left(\frac{\xi+1}{\xi}\right). \tag{C.24}$$

After multiplying by $C^2$ and inserting the fixed point value of $g_r^*$ we obtain

$$\langle\phi^N_{nrm}(x)\phi^N_{nrm}(y)\rangle_{conn} = \frac{N(N-1)\epsilon^2\mu^{2\Delta_\phi}}{8(N-2)^2}\log^2\left(\frac{\xi+1}{\xi}\right). \tag{C.25}$$

Expanding this in boundary conformal blocks we get a spectrum of boundary operators with $\hat{\Delta} = 2, 4, 6, 8\ldots$. The leading operator with $\hat{\Delta} = d \approx 2$ is the displacement operator. The first few OPE

coefficients are:

$$\text{Boundary channel, longitudinal}: \quad \mu_n = \frac{N(N-1)\epsilon^2}{8(N-2)^2}\left\{1, \frac{1}{60}, \frac{1}{1890}, \frac{1}{48048}\right\}, \quad \hat{\Delta}_n = \{2,4,6,8\}. \tag{C.26}$$

Using the `FindSequenceFunction` in `Mathematica`, we guess

$$\mu_{\hat{\Delta}=2n+2} = \frac{N(N-1)\epsilon^2}{8(N-2)^2}\frac{((2n)!)^2}{(n+1)(4n+1)!}, \quad n \geq 0, \tag{C.27}$$

which we have checked up to $\hat{\Delta} = 100$. Note that $\mu_{\hat{\Delta}}$ is positive. Further, the first operator beyond the displacement has $\mu$ suppressed by $1/60$ compared to the displacement operator. This might partly justify the truncation in section 4. The order $\epsilon$ shift in the dimension of these operators was computed in [80], from the four-point function of the tilt operator.

Now, combining the longitudinal and transverse contributions, we get:

$$\langle \phi^a_{nrm}(x)\phi^a_{nrm}(y)\rangle_{conn} = \frac{\epsilon N(N-1)}{2(N-2)}\frac{1}{(x-y)^{2\Delta_\phi}}\left[\log\left(\frac{\xi+1}{\xi}\right)\right.$$
$$\left. + \epsilon\left(\text{Li}_2(-1/\xi) - \frac{1}{N-2}\log\left(\frac{\xi+1}{\xi}\right)\left(1 + \frac{N-3}{2}\log\xi\right) - \frac{1}{4}\frac{N-3}{N-2}\log^2\left(\frac{\xi+1}{\xi}\right)\right)\right]. \tag{C.28}$$

Now,

$$\langle \phi^a_{nrm}(x)\phi^a_{nrm}(y)\rangle_{conn} = \langle \phi^a_{nrm}(x)\phi^a_{nrm}(y)\rangle - \frac{a_\sigma^2}{(4x_dy_d)^{\Delta_\phi}} \xrightarrow{\xi\to 0} \frac{N}{(x-y)^{2\Delta_\phi}}\left(1 - \frac{a_\sigma^2}{N}\xi^{\Delta_\phi}\right)$$
$$\approx \frac{N}{(x-y)^{2\Delta_\phi}}\left(1 - \frac{a_\sigma^2}{N}(1 + \Delta_\phi\log\xi + \frac{1}{2}\Delta_n^2\log^2\xi)\right). \tag{C.29}$$

Matching this to Eq. (C.28) for $\xi \to 0$, we obtain

$$\mu_\sigma = a_\sigma^2 \approx N\left(1 + \frac{\pi^2}{12}\frac{N-1}{N-2}\epsilon^2\right). \tag{C.30}$$

Note that we also reproduce $\Delta_\phi$ correctly to order $\epsilon^2$, Eq. (C.4). Using the expression for $a_\sigma^2$, the full two-point function becomes:

$$\langle \phi^a_{nrm}(x)\phi^a_{nrm}(y)\rangle = \frac{N}{(x-y)^{2\Delta_\phi}}\left[1 + \frac{\epsilon(N-1)}{2(N-2)}\log(1+\xi)\right.$$
$$\left. + \frac{\epsilon^2(N-1)}{2(N-2)}\left(-\text{Li}_2(-\xi) - \frac{1}{N-2}\log(1+\xi) - \frac{1}{4}\frac{N-3}{N-2}\log^2(1+\xi)\right)\right]. \tag{C.31}$$

Let's discuss the bulk channel decomposition of the above two-point function. The leading bulk operator (besides the identity) that contributes has dimension $\Delta = 2 + O(\epsilon^2)$. Now,

$$f_{\text{bulk}}(2,\xi) \approx \log(1+\xi) - \frac{\epsilon}{2}(\text{Li}_2(-\xi) + \log(1+\xi) + \frac{1}{2}\log^2(1+\xi)) + O(\epsilon^2). \tag{C.32}$$

So

$$\langle \phi^a_{nrm}(x)\phi^a_{nrm}(y)\rangle$$
$$= \frac{N}{(x-y)^{2\Delta_\phi}}\left[1 + \lambda_{\Delta=2}f_{\text{bulk}}(2,\xi) + \frac{\epsilon^2(N-1)}{4(N-2)}\left(-\text{Li}_2(-\xi) - \log(1+\xi) + \frac{1}{2(N-2)}\log^2(1+\xi)\right)\right], \tag{C.33}$$

with

$$\lambda_{\Delta=2} \approx \frac{\epsilon(N-1)}{2(N-2)}\left(1 + \frac{N-3}{N-2}\epsilon\right). \tag{C.34}$$

We see that in addition to the operator with $\Delta \approx 2$ an infinite series of bulk operators with $\Delta = 4, 6, 8, 10 \ldots$ contribute to the two-point function with the OPE coefficient $\lambda \sim O(\epsilon^2)$. We may write

$$\lambda_\Delta = \frac{\epsilon^2(N-1)}{4(N-2)}(\alpha_\Delta + \frac{1}{N-2}\beta_\Delta), \tag{C.35}$$

with the first few coefficients

$$\alpha_4 = \frac{1}{4}, \quad \alpha_6 = \frac{1}{36}, \quad \alpha_8 = \frac{1}{240}, \quad \alpha_{10} = \frac{1}{1400}, \tag{C.36}$$

$$\beta_4 = \frac{1}{2}, \quad \beta_8 = \frac{1}{120}, \quad \beta_{12} = \frac{1}{3780}, \quad \beta_{16} = \frac{1}{96096}. \tag{C.37}$$

Note that $\beta_\Delta$ is non-zero only for $\Delta$ - a multiple of four. Using the `FindSequenceFunction` in `Mathematica`, we guess

$$\alpha_{2n+4} = \frac{(n!)^2}{2(n+2)(2n+1)!}, \quad \beta_{4n+4} = 2\alpha_{4n+4}, \qquad n \geq 0. \tag{C.38}$$

We have checked that equation (C.38) holds up to $\Delta = 200$. We observe that $\alpha_\Delta$ and $\beta_\Delta$ are positive, supporting the conjecture in section 5. Note that for $\epsilon \to 0$ there are degeneracies in the operator spectrum. For instance, at $\Delta \approx 4$ there are two $O(N)$ singlet operators (linear combinations of $(\partial_\mu \phi^a \partial_\mu \phi^a)^2$ and $(\partial_\mu \phi^a \partial_\nu \phi^a)(\partial_\mu \phi^b \partial_\nu \phi^b)$) [82]. Thus, $\lambda_\Delta$ denotes the sum of $\lambda$'s of all operators within each degenerate manifold with a given dimension $\Delta$. For $\Delta \geq 4$, we will not be able to resolve the OPE coefficients associated with the individual operators; while we have confirmed that their sum $\lambda_\Delta$ is positive, some of the individual coefficients could be negative. Higher order calculations in $\epsilon$ would be needed to resolve these degeneracies.

We conclude this section by setting $\epsilon = 1$ in eqs. (C.23), (C.30) and comparing the result to $\mu_t$, $\mu_\sigma$ obtained with the truncated bootstrap, table 3, and with Monte-Carlo, table 2. We see that $\mu_t$ obtained this way is negative for all $N$—an unphysical result. Keeping just the $O(\epsilon)$ term in $\mu_t$ would give a positive value, but one that does not agree well with the truncated bootstrap, the Monte-Carlo or the large-$N$ expansion in $d = 3$ for $N \to \infty$. On the other hand, $\mu_\sigma$ for $N = 3$ is about 20% smaller than the Monte-Carlo value; the comparison of $\mu_\sigma$ with the truncated bootstrap for $N = 3, 4, 5$ yields a similar magnitude of deviation, see figure 4a. We note that even if the $2 + \epsilon$ expansion is not particularly useful for extracting the numerical values of $\mu_\sigma$, $\mu_t$ in $d = 3$, it still serves as a non-trivial test of the bulk positivity conjecture in section 5.

## C.2 Large $N$ expansion

This section is devoted to large-$N$ results on the normal universality class. Throughout this section $d = 3$. In Ref. [31] the correlation function $G_S = \langle \phi^a(x)\phi^a(y) \rangle$ was computed in the large-$N$ expansion to first subleading order following the methods of Ref. [42]:

$$\langle \phi^a(x)\phi^a(y) \rangle = (N-1)G_\phi(x,y) + G_\sigma(x,y) = \frac{N}{(x-y)^{2\Delta_\phi}}(h^0(\xi) + \frac{1}{N}h^1(\xi)) \tag{C.39}$$

$$= \frac{N}{(x-y)^{2\Delta_\phi}}\xi^{\Delta_\phi}\left(\frac{\mu_\sigma}{N} + \frac{1}{4}\left(1 - \frac{4}{3\pi^2 N}\right)f_{\text{bry}}(2,\xi) + \frac{1}{N}g^1(\xi)\right), \tag{C.40}$$

with

$$\mu_\sigma = 2N\left(1 + \frac{1}{N}\left(1 - \frac{4}{3\pi^2}\right) + O(N^{-2})\right),$$

$$\mu_t = \frac{1}{4}\left(1 + \frac{1}{N}\left(1 - \frac{4}{3\pi^2}\right) + O(N^{-2})\right), \tag{C.41}$$

$$h^0(\xi) = \frac{1 + 2\xi}{\sqrt{1 + \xi}}, \tag{C.42}$$

$$h^1(\xi) = \frac{8\sqrt{\xi}}{\pi^2}\left(\frac{1}{6}\frac{1 + 2\xi}{\sqrt{\xi(1+\xi)}}\log(1+\xi) - \frac{s}{3} + \mathrm{Li}_2(s) - \mathrm{Li}_2(-s) - \log s \log\frac{1+s}{1-s}\right), \qquad s = \sqrt{\frac{\xi}{1+\xi}}, \tag{C.43}$$

$$f_{\mathrm{bry}}(2,\xi) = 8\left(\frac{\xi + 1/2}{\sqrt{\xi(\xi+1)}} - 1\right), \tag{C.44}$$

$$g^1(\xi) = \frac{8}{\pi^2}\left(\frac{1}{6\sqrt{\xi(\xi+1)}}(1 - 2(2\xi+1)\log s) + \mathrm{Li}_2(1 - 1/s) + \mathrm{Li}_2(-1/s) - \log s \log(1 + s^{-1}) + \frac{\pi^2}{12}\right). \tag{C.45}$$

We can also compute the longitudinal and transverse components of the two point function:

$$\langle \phi^N(x)\phi^N(y)\rangle = \frac{1}{(x-y)^{2\Delta_\phi}}\xi^{\Delta_\phi}\left(\mu_\sigma + p^1(\xi)\right), \tag{C.46}$$

$$\langle \phi^i(x)\phi^j(y)\rangle = \frac{\delta^{ij}}{(x-y)^{2\Delta_\phi}}\xi^{\Delta_\phi}\left(\mu_t f_{\mathrm{bry}}(2,\xi) + \frac{1}{N}q^1(\xi)\right), \tag{C.47}$$

where

$$p^1(\xi) = \frac{8}{\pi^2}\left[\log s - \frac{\xi + 1/2}{\sqrt{\xi(\xi+1)}}\left(\mathrm{Li}_2(-1/s) + \mathrm{Li}_2(1 - 1/s) - \log s \log(1 + s^{-1}) + \frac{\pi^2}{12}\right)\right],$$

$$q^1(\xi) = \frac{8}{\pi^2}\left[\frac{1}{6\sqrt{\xi(\xi+1)}} - \left(\frac{2\xi + 1}{3\sqrt{\xi(\xi+1)}} + 1\right)\log s\right.$$

$$\left. + \left(1 + \frac{\xi + 1/2}{\sqrt{\xi(\xi+1)}}\right)\left(\mathrm{Li}_2(-1/s) + \mathrm{Li}_2(1 - 1/s) - \log s \log(1 + s^{-1}) + \frac{\pi^2}{12}\right)\right]. \tag{C.48}$$

We now make a few comments about the two point-functions above.

*Boundary channel.* At leading order in $1/N$, the longitudinal correlation function $\langle \phi^N(x)\phi^N(y)\rangle$ is saturated by the contribution from the identity operator, while the transverse correlation function $\langle \phi^i(x)\phi^j(y)\rangle$ is saturated by the contribution from the tilt operator (dimension $\hat{\Delta}_t = 2$). At next order in $1/N$ an infinite sequence of operators contributes to both the longitudinal and transverse correlation functions.

In the transverse correlation function's boundary OPE, operators of odd dimension $\hat{\Delta} = 5, 7, 9, 11\ldots$ appear at next order in $1/N$. Note that this is consistent with our assumed form of the boundary operator spectrum (2.17). The OPE coefficients of the first few are given by:

$$\text{Transverse}: \qquad \mu_{\hat{\Delta}=5} = \frac{1}{450\pi^2 N}, \quad \mu_{\hat{\Delta}=7} = \frac{1}{7840\pi^2 N}, \quad \mu_{\hat{\Delta}=9} = \frac{1}{145152\pi^2 N}. \tag{C.49}$$

We have used the `FindSequenceFunction` in `Mathematica` to guess the general form of the sequence above

$$\mu_{\hat{\Delta}=2n+3} = \frac{n(n+1)}{3 \cdot 2^{4n-2}\pi^2(2n+1)(2n+3)^2 N}, \quad n \geq 1. \tag{C.50}$$

which we have checked up to $\hat{\Delta} = 100$. We don't know whether any of these operators are degenerate to leading order in $1/N$ (if so, the OPE coefficient reported is the sum of OPE coefficients of all the operators in the degenerate multiplet.) As expected from unitarity, the OPE coefficients are positive. Note that $\mu_{\hat{\Delta}=5}$ is numerically suppressed by three orders of magnitude compared to $\mu_{\rm t}$ (in addition to the $1/N$ suppression), potentially justifying the truncation in section 4.

In the longitudinal correlation function's boundary OPE, boundary operators of odd dimension $\hat{\Delta} = 3, 5, 7, 9, 11 \ldots$ appear at next order in $1/N$. This is again consistent with the assumption (2.17). The first of these operators with $\hat{\Delta} = 3$ is the displacement operator. The OPE coefficients of the first few are given by:

$$\text{Longitudinal}: \quad \mu_{D,\hat{\Delta}=3} = \frac{4}{9\pi^2}, \quad \mu_{\hat{\Delta}=5} = \frac{1}{225\pi^2}, \quad \mu_{\hat{\Delta}=7} = \frac{9}{78400\pi^2}, \quad \mu_{\hat{\Delta}=9} = \frac{1}{254016\pi^2}. \tag{C.51}$$

`FindSequenceFunction` guesses the following expression, which we have checked up to $\hat{\Delta} = 100$:

$$\mu_{\hat{\Delta}=2n+1} = \frac{n^2}{2^{4n-6}\pi^2(4n^2-1)^2 N}, \quad n \geq 1. \tag{C.52}$$

Again, if any degeneracy of operator dimensions is present at $N = \infty$, we are not able to resolve it here. The OPE coefficients are positive as expected. Note that $\mu_{\hat{\Delta}=5}$ is down by a factor of 100 compared to $\mu_D$ again potentially justifying the truncation in section 4.

*Bulk channel.* We now decompose the correlator Eq. (C.39) in terms of bulk conformal blocks. At leading order in $N$, starting from the $h^0(\xi)$ term in Eq. (C.39), we find contributions from bulk O($N$) singlet operators of even dimensions $\Delta = 2, 4, 6, 8 \ldots$. The first few coefficients are

$$\text{Singlet}: \quad \lambda_{\Delta=2} = \frac{3}{2}, \quad \lambda_{\Delta=4} = \frac{3}{8}, \quad \lambda_{\Delta=6} = \frac{37}{560}, \quad \lambda_{\Delta=8} = \frac{135}{9856}, \quad \lambda_{\Delta=10} = \frac{329}{109824}. \tag{C.53}$$

Using `FindSequenceFunction` we obtain

$$\lambda_{\Delta=2n+2} = \frac{(8n^2 + 4n - 3)((2n)!)^4}{2^{2n+1}(n+1)(2n-1)(n!)^4(4n)!}, \quad n \geq 0, \tag{C.54}$$

which we have checked up to $\Delta = 200$. The coefficients (C.54) are positive for all $n$ supporting the conjecture in section 5. Note that while the OPE coefficients don't fall off as rapidly with increasing $\Delta$ as in the boundary channel, $\lambda_{\Delta=10} \approx 0.003$ is already quite small, potentially justifying the truncation in section 4. Importantly, the first two bulk O($N$) singlet operators of dimensions $\Delta = 2$ and $\Delta = 4$ are non-degenerate in the $N = \infty$ limit. However, the higher lying operators are degenerate and the coefficients $\lambda$ above should be understood as the sum of OPE coefficients of all operators in each degenerate multiplet. In general, we will not be able to resolve the individual OPE coefficients of each operator in the multiplet for $\Delta \geq 8$ (i.e. it could still be possible that some of these coefficients are positive and some are negative, while their sum is positive). However, for $\Delta = 6$, we will be able to resolve the (two-fold) degeneracy and show that the individual coefficients are positive.

Let's briefly discuss some of the bulk singlet operators in the O($N$) model. If we use the non-linear $\sigma$-model formulation of the O($N$) model,

$$L = \frac{1}{2}\partial_\mu \phi^a \partial_\mu \phi^a + \frac{i\lambda}{2}\left(\phi^a \phi^a - \frac{1}{g}\right) \tag{C.55}$$

with $\lambda(x)$ - a Lagrange multiplier,[19] then one family of bulk singlet operators is given by $\lambda^k$, $k = 1, 2, 3, \ldots$, which have dimension $\Delta_k = 2k$ at $N = \infty$. The lowest two primaries $\lambda, \lambda^2$ are non-degenerate and have approximate scaling dimensions:

$$\Delta_2 = 2 - \frac{32}{3\pi^2 N}, \quad \Delta_4 = 4 - \frac{64}{3\pi^2 N}. \tag{C.56}$$

However for $k \geq 3$ one can replace some number of $\lambda$'s in $\lambda^k$ by two derivatives. For instance, at $\Delta = 6$, in addition to $\lambda^3$, we also have the operator $\lambda\partial^2\lambda$. Thus, we have at least two primaries with $\Delta \approx 6$, and the $1/N$ corrections to their scaling dimensions are known [83]:

$$\Delta_6^{(1)} = 6 - \frac{32}{\pi^2 N}, \quad \Delta_6^{(2)} = 6 - \frac{64}{3\pi^2 N}. \tag{C.57}$$

For $k \geq 4$ there are more than two Lorentz singlet primaries that can be formed out of $\lambda$ and its derivatives and the dimensions of two of these are known to $O(1/N)$, however, we won't need them below [83].

In principle, in $d = 3$ there is yet another operator with $\Delta = 6$ at $N = \infty$, schematically: $\mathcal{O}_6^{(3)} = (\partial_\mu \phi^a \partial_\nu \phi^a)(\partial_\mu \phi^b \partial_\nu \phi^b)$. At $N = \infty$, in general dimension $d$ we expect this operator to have dimension $\Delta = 2d$, which becomes $\Delta = 6$ in $d = 3$. However, repeating the calculation of $\langle \phi^a(x)\phi^a(y) \rangle$ at $N = \infty$ in arbitrary $d$, we find no operator of dimension $\Delta = 2d$ in the bulk channel (instead, we find only operators of even integer dimension). Thus, we conclude that the OPE coefficient $\lambda_{\Delta=6}^{(3)}$ associated with $\mathcal{O}_6^{(3)}$ is suppressed at $N = \infty$.

With the above remarks in mind, we use the $h^1(\xi)$ term in Eq. (C.39) to compute the $1/N$ corrections to $\lambda_{\Delta=2,4}$ and to resolve the individual $\lambda_{\Delta=6}^{(1)}$ and $\lambda_{\Delta=6}^{(2)}$ associated with operators (C.57) to $O(1)$ in $N$. Inserting the dimensions (C.56), (C.57) into the bulk conformal blocks, expanding in $1/N$ and matching to $h^1(\xi)$ we obtain:

$$\lambda_{\Delta=2} = \frac{3}{2} + \frac{44}{3\pi^2 N}, \quad \lambda_{\Delta=4} = \frac{3}{8} + \frac{32}{3\pi^2 N}, \quad \lambda_{\Delta=6}^{(1)} = \frac{3}{80} + O(1/N), \quad \lambda_{\Delta=6}^{(2)} = \frac{1}{35} + O(1/N). \tag{C.58}$$

Resolving the individual OPE coefficients for $\Delta \geq 8$ would require knowing $\langle \phi^a(x)\phi^a(y) \rangle$ to yet higher order in $1/N$ (as well as knowing all the scaling dimensions of all the operators in the degenerate multiplet to higher order in $1/N$).

Let us conclude with a comment on the consistency of these results with conformal representation theory. The primary operator $\phi^a$ has the free scaling dimension $\Delta_\phi = 1/2$ at leading order in $1/N$. It is well known that both the bulk and the boundary OPEs for free fields are constrained (see *e.g.* [7]): the only scalar allowed in the bulk has dimension $2\Delta_\phi$, while in the boundary channel only $\widehat{\Delta} = \Delta_\phi$ and $\Delta_\phi + 1$ are possible. The resolution of the tension lies in the nature of the boundary state, which is not normalizable in the $N \to \infty$ limit. Specifically, the one-point functions of the (appropriately normalized) operators $\lambda^k$ grow with $N$, and offset the decay of the three-point functions $\langle \phi\phi\lambda^k \rangle$.

## C.3  $4 - \epsilon$ expansion

The most recent computation of the two-point function of $\phi$ in $4 - \epsilon$ expansion was performed in Ref. [13]. The correlator was computed to order $\epsilon$, and the CFT data appearing in the $O(N)$ singlet combination $G_S$ were explicitly extracted. We report the results which are important for the present work, and we discuss the mixing problem in the bulk channel, which was not solved in [13].

---

[19]Not to be confused with OPE coefficients $\lambda$.

In the boundary channel, the spectrum includes the identity, the tilt, the displacement, and a tower of operators which, at leading order, have integer dimensions $\widehat{\Delta} = 6, 7 \ldots$ More precisely, the identity is the only operator with an OPE coefficient of order $1/\epsilon$:

$$\mu_\sigma(N) = \frac{4(N+8)}{\epsilon} \left( 1 - \frac{N^2 + 31N + 154}{(N+8)^2} \epsilon \right). \tag{C.59}$$

To order $O(\epsilon^0)$, the longitudinal connected correlator is saturated by the displacement, while a tower of operators with even dimensions $\widehat{\Delta} = 6, 8 \ldots$ appear at order $\epsilon$. From the transverse correlator, we learn that the tilt is the only primary appearing at zeroth order in the boundary OPE of $\phi^i$. Its OPE coefficient is

$$\mu_{\mathrm{t}}(N) = \frac{1}{3} \left( 1 - \epsilon \frac{N+9}{6(N+8)} \right). \tag{C.60}$$

At order $\epsilon$ a tower of operators appear, which have odd scaling dimensions $\widehat{\Delta} = 7, 9 \ldots$ Like in the other perturbative limits considered in this appendix, the gaps in the boundary channel are consistent with the assumptions made in eq. (2.17).

Moving on to the bulk channel, the leading order scaling dimensions of the bulk primaries appearing in the OPE is given by $\Delta_n = 2 + 2n$, $n \in \mathbb{N}$. This is easily understood from free theory in $d = 4$. Following Ref. [13], we define rescaled OPE coefficients for the operators above the identity, as follows:[20]

$$\bar{\lambda}_\Delta = \lambda_\Delta \frac{\Gamma\left( \Delta + 1 - \dfrac{d}{2} \right)}{\Gamma\left( \dfrac{\Delta}{2} \right) \Gamma\left( \dfrac{\Delta}{2} + 2 - \dfrac{d}{2} \right)}, \qquad \Delta \neq 0. \tag{C.61}$$

Let us expand the scaling dimensions and the OPE coefficients in powers of $\epsilon$ as follows:

$$\bar{\lambda}_n = \frac{1}{\epsilon} \bar{\lambda}_{n,-1} + \bar{\lambda}_{n,0} + \ldots, \qquad \Delta_n = 2 + 2n + \gamma_{n,1}\epsilon + \gamma_{n,2}\epsilon^2 + \ldots. \tag{C.62}$$

The leading order OPE coefficients are

$$\langle \bar{\lambda}_{0,-1} \rangle = \frac{4(N+8)}{N}, \qquad \langle \bar{\lambda}_{n,-1} \rangle = \frac{8(N+8)}{N}, \ n > 0. \tag{C.63}$$

They are positive, in agreement with the conjecture which makes the use of SDPB possible. However, since in four dimensions $\partial$ and $\phi$ have the same engineering dimension, primaries in the free theory are increasingly degenerate. The brackets in the previous equation precisely denote the sum over all primaries which are degenerate at the free fixed point, e.g.[21]

$$\langle \bar{\lambda}_{n,-1} \rangle = \sum_{i=\mathrm{degenerate}} \bar{\lambda}_{n,-1}^{(i)}. \tag{C.66}$$

---

[20]The relation between our definition and that of Ref. [13] involves a numerical factor: $\bar{\lambda}_\Delta = 4a_\Delta^{[13]}/(N(2+\epsilon))$. In the conventions chosen here, the external operators are unit-normalized.

[21]Eq. (C.61) can be used to translate these averages to averages of $\lambda$'s. One must be careful, since the factor that relates the two quantities depends on $\Delta$. For instance, defining

$$K(\Delta, \epsilon) = \frac{\Gamma\left( \dfrac{\Delta}{2} \right) \Gamma\left( \dfrac{\Delta + \epsilon}{2} \right)}{\Gamma\left( \Delta - 1 + \dfrac{\epsilon}{2} \right)}, \tag{C.64}$$

one has

$$\langle \lambda_{n,0} \rangle = \langle \bar{\lambda}_{n,0} \rangle K(2+2n, 0) + \langle \bar{\lambda}_{n,-1} \gamma_{n,1} \rangle K^{(1,0)}(2+2n, 0) + \langle \bar{\lambda}_{n,-1} \rangle K^{(0,1)}(2+2n, 0), \tag{C.65}$$

where the upper indices on $K$ denote partial derivatives with respect to the two arguments of $K(\Delta, \epsilon)$.

The first two-fold degeneracy appears for $n = 2$, and we will be able to solve for the two individual coefficients. However, let us first discuss the general case. From the two-point function up to $O(\epsilon)$, Ref. [13] found

$$\frac{\langle \bar{\lambda}_{n,-1} \gamma_{n,1} \rangle}{\langle \bar{\lambda}_{n,-1} \rangle} = 6 \frac{n^2 - 1}{N + 8}, \qquad \frac{\langle \bar{\lambda}_{n,-1} (\gamma_{n,1})^2 \rangle}{\langle \bar{\lambda}_{n,-1} \rangle} = \left( 6 \frac{n^2 - 1}{N + 8} \right)^2, \quad n \geq 0. \tag{C.67}$$

Now, *if* the individual OPE coefficients of the degenerate operators are positive, then eq. (C.67) admits only two solutions. The first option is that no degeneracy is lifted at this order: for each $n$, all scaling dimensions are still identical at order $\epsilon$. The second option is that, at each value of $n$, all $\bar{\lambda}_{n,-1}^{(i)}$'s vanish except for one.[22] In fact, a direct computation shows that at least the two primaries of dimension $\Delta_2 = 6 + O(\epsilon)$ are not degenerate at order $\epsilon$:

$$\gamma_{2,1}^{(1)} = -\frac{12}{N + 8}, \qquad \gamma_{2,1}^{(2)} = \frac{18}{N + 8}. \tag{C.68}$$

Furthermore, in the case of a two-fold degeneracy the positivity assumption is not necessary: the only solution to eqs. (C.63) and (C.67) for $n = 2$ is

$$\bar{\lambda}_{2,-1}^{(1)} = 0 \ , \qquad \bar{\lambda}_{2,-1}^{(2)} = \frac{8(N + 8)}{N}, \qquad \gamma_{2,1}^{(2)} = \frac{18}{N + 8}, \tag{C.69}$$

in agreement with eq. (C.68). Hence, we confirm that the $\bar{\lambda}$'s are positive in the only case where we can solve the degeneracy completely. The anomalous dimension $\gamma_{2,1}^{(2)}$ corresponds to the operator $\phi^6$. It is interesting to notice that this result is easily confirmed by a direct computation of $\bar{\lambda}_{2,-1}^{(1)}$. Let us sketch the argument. The associated operator is of the schematic form $\phi^2 \Box \phi^2$, and it does not mix with $\phi^6$ at order one. At leading order, the one-point functions are obtained by evaluating the fields on the classical solution, which is $\langle \phi \rangle \sim \epsilon^{-1/2}$—see eq. (C.59). Hence, $\langle \phi^2 \Box \phi^2 \rangle = O(\epsilon^{-2})$. On the other hand, twist[23] four operators are known not to appear in the OPE of the fundamental field up to order $\epsilon^2$, despite the naive loop counting. This was first noticed in the context of boundary bootstrap in [5], then confirmed by an $O(\epsilon^2)$ computation in [84]. Furthermore, eq. (C.69) leads to a prediction for the coefficient of the three-point function $\langle \phi \phi \mathcal{O}_3 \rangle$, where $\mathcal{O}_k$ is the unit normalized version of $\phi^{2k}$. Indeed, the one-point function of $\phi^6$ is easily computed at leading order from eq. (C.59):

$$a_{\phi^6} = \left( \frac{4(N + 8)}{\epsilon} \right)^3. \tag{C.70}$$

The three-point coefficient is then computed by dividing out $\lambda_2$ by $a_{\phi^6}$ and compensating for the normalization of the operator. In general, one has, in free theory,

$$\phi^{2k}(x) \phi^{2k}(y) \sim \frac{M(k)}{(x - y)^{4k\Delta_\phi}} + \dots, \quad x \to y, \tag{C.71}$$

where

$$M(k) = 2^{2k} k! \left( \frac{N}{2} \right)_k. \tag{C.72}$$

---

[22]If there are degenerate operators with $\gamma_{n,1} = 6(n^2 - 1)/(N + 8)$, all those are obviously allowed to have non-vanishing OPE coefficient.

[23]The twist is defined as the scaling dimension minus the spin.

Hence, recalling the relation (C.61) between $\bar\lambda$'s and $\lambda$'s, the three-point function coefficient is

$$c_{\phi\phi\mathcal{O}_3} = \frac{\lambda_{2,-1}^{(2)}}{\epsilon\, a_{\phi^6}} \sqrt{M(3)} = \frac{1}{12(N+8)^2} \sqrt{\frac{3(N+2)(N+4)}{N}} \epsilon^2. \tag{C.73}$$

It is noteworthy that a two-loop result about the bulk CFT can be obtain from an $O(\epsilon)$ computation in a BCFT. This result was known in the Ising model case, $N = 1$ [76].

What about the higher dimensional primaries? The degeneracy grows, as one can check for instance with the computation of a character. Therefore, we cannot conclude from eq. (C.67) that all the operators appear with non-negative OPE coefficient. However, it is tempting to reverse the logic and see what we can learn from the assumption of positivity. The averaged anomalous dimension in eq. (C.67) equals that of the operator $\mathcal{O}_{n+1} \propto \phi^{2+2n}$ [13]. Moreover, it can be shown that this operator does not mix with any other at $O(\epsilon^0)$ [85]. Finally, besides positivity, we need to assume that no other operator is degenerate with $\mathcal{O}_k$ at order $\epsilon$. If this is the case, $\mathcal{O}_k$ is the only operator with a non-vanishing $\lambda$, and we can extract its coefficient in the $\phi \times \phi$ OPE:

$$c_{\phi\phi\mathcal{O}_1}^{\mathrm{conj}} = \sqrt{\frac{2}{N}}, \tag{C.74a}$$

$$c_{\phi\phi\mathcal{O}_k}^{\mathrm{conj}} = \frac{8}{2^k k^2 (2k-2)!(N+8)^{k-1} N} \sqrt{(k!)^5 \left(\frac{N}{2}\right)_k} \epsilon^{k-1}, \quad k > 1. \tag{C.74b}$$

This prediction, which we denoted as conjectural, corresponds to loop computations of arbitrarily high order in the bulk CFT. Eq. (C.74) is a true formula, rather than a conjecture, for $k = 1, 2, 3$, and we checked it for $k = 1, 2$ [5, 77] at any $N$, and 3 at $N = 1$ [76]. It would be intersting to check it at higher order, to confirm the positivity of the OPE coefficients.[24]

We know that one of the two operators at $n = 2$ has vanishing bulk OPE coefficient at order $\epsilon^{-1}$, and that, if positivity has a chance, infinitely many others do as well. It is then worth proceeding to the next order to check if the $\bar\lambda$'s are non-negative, at least up to degeneracies. Ref. [13] found the following:

$$\langle\bar\lambda_{0,0}\rangle = -\frac{3N^2 + 106N + 536}{N(N+8)}, \tag{C.75}$$

$$\langle\bar\lambda_{n,0}\rangle = -\frac{4}{N}\left[46 + 2N - \frac{60}{N+8} - 12n + (N+8)\left(\frac{3}{2n} + 2H_{n-1}\right)\right], \quad n > 0, \tag{C.76}$$

where

$$H_n = \sum_{k=1}^{n} \frac{1}{k}. \tag{C.77}$$

Although $\langle\bar\lambda_{n,0}\rangle$ is positive for large enough $n$, it is negative for the first few values. In particular, $\langle\bar\lambda_{2,0}\rangle$ is negative for any value of $N$. Therefore, the only chance for positivity is that the negative contribution comes from the OPE coefficient of $\mathcal{O}_{n+1}$ (and of operators degenerate with it at order $\epsilon$).

---

[24]If the one-point functions of the other primaries does not vanish at leading order, we obtain as a by-product that the three-point function coefficients $c_k$ of these operators all have to start at the next order with respect to the naive loop counting. It would be interesting to perform this computation. Notice that computing the one-point functions only require diagonalizing the operators in the free theory, which is a simpler exercise than computing the three-point function.

For the leading degeneracy, $n = 2$, this question can be settled. Indeed, from the correlator computed in [13] one can also extract the following sum:

$$\langle \bar{\lambda}_{n,-1} \gamma_{n,2} \rangle + \langle \bar{\lambda}_{n,0} \gamma_{n,1} \rangle. \tag{C.78}$$

The second order anomalous dimension for the operator $\mathcal{O}_{n+1}$ is known [86]:

$$\gamma_{\mathcal{O}_{n+1},2} = -\frac{n+1}{(N+8)^3} \left[ n(34(n-1)(N+8) + 11N^2 + 92N + 212) - \frac{1}{2}(13N + 44)(N+2) \right]. \tag{C.79}$$

Together, with eqs. (C.68), (C.69), and (C.76), we can solve for the two contributions to the OPE coefficients:

$$\bar{\lambda}_{2,0}^{(1)} = 0, \qquad \bar{\lambda}_{2,0}^{(2)} = -\frac{19N^2 + 328N + 1168}{N(N+8)}. \tag{C.80}$$

Hence, the positivity assumption is still safe. It would be interesting to construct explicitly the operator associated to $\bar{\lambda}_2^{(1)}$, and compute the one-point function. If it is of the naive order $\epsilon^{-2}$, then the three-point coefficient has to be $O(\epsilon^3)$, *i.e.* even more suppressed than currently known from the four-point function [5, 84].

As for the heavier primaries, with $n > 2$, we do not have enough information at this order to solve the mixing problem.[25]

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
