# Peer review of "The extraordinary boundary transition in the 3d O(N) model via conformal bootstrap"

_SciPost Physics_

## Round 1 · Referee Report · Slava Rychkov (Referee 1) · 2022-2-6

Report

Recently, Metlitski proposed a new boundary universality class for the O(N) model happening for the bulk coupling fixed to the critical value, and for the boundary coupling sufficiently large. This class, dubbed "extraordinary log", should be realized for 2<N<N_c where the critical value N_c is not yet very well ascertained. MC simulations suggest the new transition is realized for N=3,4, making this phenomenologically interesting.

Metlitski showed that N_c can be determined by studying the "normal" boundary phase transition in the O(N) model, where one adds a boundary magnetic field breking O(N) invariance on the boundary explicitly to O(N-1). Some boundary CFT quantity C_t related to this transition determines N_c.

The paper under review is devoted to the numerical bootstrap studies of the normal transition, and to extraction of the quantity C_t.

The paper performs this extraction using two techniques: truncation method a la Gliozzi, and positive bootstrap. Both methods have some caveats, of different origin, so they complement each other. The paper gives a very honest assessment of the theoretical uncertainties associated with both methods. Truncation method shows N_c>4, and positive bootstrap gives a slightly weaker result N_c>3. Their results are this nicely compatible with MC simulations.

The paper is generally very well written and I recommend it for publication. I have a few minor remarks. I leave it to the authors to consider if they want to incorporate any changes in reaction to these remarks.

  • Table 1: paper 2111.12093 has now a rigorous estimate of Delta_{\epsilon'} for N=1. They might consider mentioning it.

  • Table 1 caption: "values in regular font refer to single correlator bootstrap". Is this true for all regular font results in this table? At least some of these results seem to me multiple correlator bootstrap results with conformal data extracted by the Extremal Functional Method, with error determined by scanning over a sample of allowed points from the island (hence not fully rigorous).

  • p.11: I found the logic of the second paragraph of section 3 hard to follow. The second sentence of this paragraph ("At this length scale, we expect the system to flow to the normal fixed point.") sounds strange. How can the system approach normal fixed point if they later argue that for N<N_c it actually approaches log-extraordinary phase (which is strictly different)?

Moreover, they don't even seem to need this sentence for the validity of the rest of this section. They only use a weaker claim that the endpoint of RG flow of the lattice model (1.1) can be reached from the action (3.4). That's reasonable to me because (1.1) and (3.4) have the same symmetry. They don't seem to use that the RG flow of (1.1) passes anywhere near the normal fixed point without extra term in (3.4).

I'd be grateful if the authors double check this issue.

  • footnote 13: I am not sure how to interpret the first sentence. Do the authors mean perhaps that they could assume a gap above the displacement? Even then, I don't see how a scan can change what they are doing. Scans over OPE coefficients don't help in the positive bootstrap. Scans over ratios of OPE coefficients do, if the operator in question couples to several external states.
  • validity: high
  • significance: high
  • originality: high
  • clarity: high
  • formatting: excellent
  • grammar: excellent

Author:  Ilya Gruzberg  on 2022-03-31  [id 2345]

(in reply to Report 1 by Slava Rychkov on 2022-02-06)

Dear Editors,

We would like to thank the referee for his careful assessment of the manuscript and his useful comments. We made a few changes to the text, which we detail below, in correspondence with the four remarks by the referee.

  1. We have modified the entry in Table 1 to refer to the new result for $N = 1$.

  2. We changed the last line of the caption for Table 1 to `values in regular font are from bootstrap techniques where the uncertainty is not rigorous'.

  3. Regarding the sentence on page 11. We have replaced the sentence with the following: ``At this length-scale, we expect the renormalization group trajectory to pass close to the normal fixed point.'' We hope this clarifies the meaning of the paragraph. We believe that making this point is important, because while it is true that Eqs. (1.1) and (3.4) have the same symmetry, what makes the model in Eq.(3.4) useful is that it is weakly coupled at intermediate scales, where the fate of the RG flow is decided. We also added the footnote 9, to make the difference between the normal fixed point and the extra-ordinary one precise: there is a set of decoupled Goldstone bosons in the latter. They don't affect bulk correlation functions at the fixed point. They are instead crucial along the flow, because they are responsible for the logarithmic decay of correlations along the boundary. This is explained in the paragraph above Eq. (3.8).

  4. Regarding the (former) footnote 13. What we had in mind involved scanning over a restricted range of values for the OPE coefficient of the displacement. This would express the bounds on $\alpha$ as a function of the range in question. However, this idea is only useful if (approximate) bounds on the size of the OPE coefficient of the displacement are known. Since this is, at the moment, an abstract thought, we decided to remove the footnote altogether.

We hope that with these modifications the paper can be accepted for publication.

J. Padayasi, A. Krishnan, M. A. Metlitski, I. A. Gruzberg, M. Meineri

---

## Editorial Decision

resubmitted